# GUARANTEED BOUNDING MESHES EXTRACTION FROM NEURAL IMPLICIT SURFACES VIA NEURAL NETWORK VERIFICATION

## ABSTRACT

Geometric queries on neural implicit surfaces, such as ray tracing and collision detection, present a significant challenge since they require explicit spatial reasoning over neural networks. This work addresses this challenge by connecting these geometric queries to neural network verification problems. Inspired by the state-of-the-art neural verification tools, we propose a new framework utilizing linear bound propagation-based verifiers to solve these queries in real time, enabling applications such as real-time rendering and physics simulation with soundness guarantees. Instead of naively running neural network verifiers on-the-fly, we first classify a 3D input domain into multiple regions of interest, which can then assist in subsequent verifications. We achieve this objective by constructing explicit bounding volumes and then leveraging linear bounds generated by SOTA neural network verifiers to guide the generation of *sound piecewise linear bounding meshes*. In this paper, we propose Guaranteed Inner-and-Outer Meshes (GIOM), which can serve as bounding volumes and merge seamlessly with existing explicit geometry processors to accelerate queries on neural implicits. As tight and *sound* bounding meshes, GIOM enables accelerated neural SDF queries without sacrificing quality. With GIOM, we develop accelerated neural implicit ray casting, collision detection, and constructive solid geometry methods (CSG), achieving up to a 300% speedup in real-time rendering, a 500% speedup in physics simulation, and an optimization-free neural CSG procedure. Experiments show that GIOM significantly outperforms existing methods in the speed-quality trade-off.

## 1 INTRODUCTION

Neural implicit surfaces, particularly those represented as signed distance functions (SDFs), provide compact, high-fidelity representations of complex 3D geometry. Their continuous and resolution-independent nature makes them ideal for tasks such as shape modeling, completion, and generative design. However, their implicit formulation, typically as multi-layer perceptrons (MLPs), poses challenges for geometry queries central to computer graphics and simulation, including ray intersection, collision detection, and boolean operations (e.g., constructive solid geometry). These tasks often require explicit spatial reasoning and, in many cases, formal guarantees on intersection, enclosure, or safe clearance (Sharp and Jacobson, 2022; Liu et al., 2024b; Marschner et al., 2023a), which implicit neural representations inherently struggle to satisfy.

To address these challenges, recent works (Liu et al., 2024a;c; Sharp and Jacobson, 2022; Wang et al., 2023b) have focused on extracting bounding volumes from neural SDFs to accelerate downstream tasks. However, these approaches trade off either soundness or tightness, hindering their utility in safety-critical and performance-intensive settings. Our key insight is that deriving formal guarantees over neural SDFs is closely connected to the field of neural network (NN) verification. In particular, bound verification techniques (Zhang et al., 2018; Gehr et al., 2018; Singh et al., 2018; Weng et al., 2018; Wong and Kolter, 2018a; Dvijotham et al., 2018; Wang et al., 2018) have demonstrated the ability to compute tight, certified bounds on NN outputs in a highly scalable and efficient manner, providing a compelling foundation for deriving tight and certifiable bounds over neural implicit surfaces. In this work, we propose **G**uaranteed **I**nner-and-**O**uter **M**eshes (GIOM), a novel algorithm that synergizes the rigor of neural network verification with the performance demands of

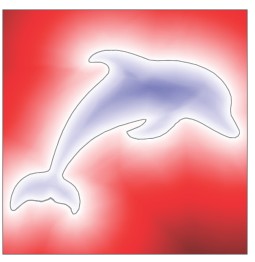 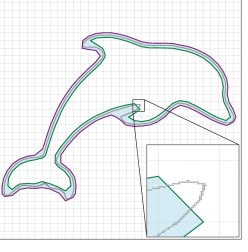 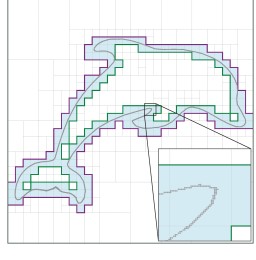 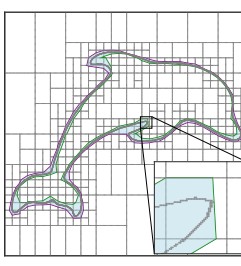

(a) Original Neural SDF     (b) Adaptive Shells     (c) Spelunking the Deep     (d) GIOM (Ours)

Figure 1: **Comparison of bounding-shell methods for Neural SDFs at equal granularity. Adaptive Shells** (Wang et al., 2023b) use *SDF dilation and erosion with marching squares* to produce relatively tight shells but lack correctness guarantees (see intersecting isosurfaces in Fig. 1b); **Spelunking the Deep** (Sharp and Jacobson, 2022) employs a *KD-tree with affine arithmetic*, yielding loose, non-smooth bounds (Fig. 1c); our **Guaranteed Inner-and-Outer Meshes (GIOM)** method exploits CROWN (Zhang et al., 2018), one state-of-the-art network verification technique to deliver tight, smooth bounds with provable correctness (Fig. 1d).

real-time graphics applications. Our work utilizes CROWN Zhang et al. (2018), a representative verification method with linear bound propagation, and organizes the input domain into a spatial hierarchical grid to progressively and adaptively refining voxel enclosures. The 3D planes derived from linear bound coefficients allow us to subsequently slice voxels and extract volumetric meshes that tightly encloses the surface. Unlike prior voxel-based methods Sharp and Jacobson (2022) only addressed soundness (the inner bounding volume lies within and the outer bounding volume contains the surface), GIOM produces geometry-aware shells that are directly compatible with spatial acceleration structures. More importantly, GIOM offers adjustable tightness-efficiency trade-offs while ensuring soundness, enabling the same core algorithm to support multiple downstream tasks ranging from fast ray tracing to certified collision detection.

We validate GIOM across three tasks. For real-time rendering, GIOM achieves frame rates 3× faster than Wang et al. (2023b) and yields 13 dB peak signal-to-noise-ratio (PSNR) over 0-level marching cubes, eliminating common artifacts. In collision detection, we provide tighter bounding volumes and more efficient queries, outperforming heuristic-based alternatives. For constructive solid geometry (CSG) operations, our shell-based framework supports robust boolean operations without the large approximation errors introduced by traditional min/max operations or the excessive overhead spent on training a new neural implicit Marschner et al. (2023a). In summary, our main contributions are:

- We connect *rigorous bounding techniques* from NN verification with the *soundness and performance demands* of visual computing with neural implicits. By formalizing this connection in Section 3.1, we enable cross-domain insights for reliable, high-performance queries on neural implicit surfaces.
- We introduce GIOM, the first method to leverage sound linear bounds on neural networks to construct tight, scalable bounding meshes tailored for 3D graphics applications. We also show GIOM enables three challenging downstream tasks, rendering, collision detection, and CSG, each of which benefits from tightness, generality, efficiency, and formal guarantees.
- We demonstrate that GIOM provides a 3× speed-up and a 5 dB PSNR improvement in rendering, a 5× speed-up in collision detection without loss of authenticity, and a 10× reduction in CSG distance field reconstruction error with minimal overhead.

## 2 RELATED WORK

**Neural Implicit Representations.** Neural implicit representations model geometry as continuous functions parameterized by NNs. SDF-based methods (Park et al., 2019; Sitzmann et al., 2020; Wang et al., 2021a; Takikawa et al., 2021) define surfaces as zero level sets, while occupancy-based approaches (Mescheder et al., 2019; Tang et al., 2021) classify points as interior or exterior. Other works employ explicit primitives (Chen et al., 2020; Deng et al., 2020; Tretschk et al., 2021; Esposito et al., 2025), volumetric fields Mildenhall et al. (2020); Zhang et al. (2020); Wang et al. (2023a), and hybrid approaches Martel et al. (2021); Guédon and Lepetit (2024) for view synthe-

sis. Neural SDFs offer compact and expressive geometry representations and are widely used in applications such as scene reconstruction Wang et al. (2021a; 2023a); Zhang et al. (2024; 2020); Müller et al. (2022); Fridovich-Keil and Yu et al. (2022), 3D modeling Chen and Zhang (2019); Li et al. (2022); Novello et al. (2023); Yang et al. (2021), and collision detection Koschier et al. (2017); Macklin et al. (2020). However, they are costly to query, challenging to edit, and inefficient for ray tracing or certified collision avoidance. To mitigate this, some acceleration techniques have been proposed. NGLOD (Takikawa et al., 2021) sparsely voxelizes space and uses compact MLPs; NeRF variants (Reiser et al., 2023; Müller et al., 2022; Martin-Brualla et al., 2021; Barron et al., 2021; Fridovich-Keil and Yu et al., 2022) exploit multi-resolution hash grids; da Silva et al. (2022) leverages nested neural SDFs; Adaptive Shells (Wang et al., 2023b) restricts sampling to narrow bands via dilation and erosion with marching cubes (Lorensen and Cline, 1987). For collision detection, affine arithmetic (De Figueiredo and Stolfi, 2004) has been applied (Sharp and Jacobson, 2022; Liu et al., 2024b) but struggles with the non-convex nature of neural networks. CSG-nSDF (Marschner et al., 2023a) trains neural SDFs to encode constructive solid geometry.

**Certified Bound Extraction and Mesh Enclosures.** Standard surface extraction methods such as marching cubes generate triangle meshes for the zero level set, but few address the challenge of computing certified inner and outer meshes that provably bound the isosurface. Differentiable variants of marching cubes (Liao et al., 2018; Remelli et al., 2020) learns explicit meshes, but only for the zero level set and without bounding guarantees. In special cases like ReLU networks, the level set is polyhedral and could, *in theory*, be triangulated exactly. However, this is computationally expensive and does not extend to more expressive architectures (Lei and Jia, 2020). Other approaches, such as Wang et al. (2023b), approximate inner and outer shells via dilation and erosion combined with marching cubes, but these are primarily intended to accelerate sampling and offer no guarantees, often resulting in unsound or visually incorrect renderings (see Figure 1, 5). Although errors may be reduced by increasing marching cubes resolution, adjusting dilation/erosion parameters, or moving bounding-mesh vertices along SDF normals, these heuristics introduce trade-offs in memory consumption, bounding-mesh tightness, and risks of self-intersection in thin or complex regions. More importantly, none of them can provide a theoretical guarantee that the implicit surface lies strictly between the bounding meshes.

**Bound Computation.** Bound propagation is a core technique in NN verification, enabling formal guarantees for tasks related to safety and robustness analysis (Li et al., 2025b; Yang et al., 2024; Serry et al., 2025; Li et al., 2025a; Chen et al., 2024). Among these methods, Interval Bound Propagation (IBP) (Gowal et al., 2019; Moore et al., 2009) has been adopted for tasks regarding certified querying and collision detection (Sharp and Jacobson, 2022; Liu et al., 2024b) due to its fast manner of forward-propagating bounds throughout the layer of a network. However, IBP tends to produce conservative bounds and ignores the structure of the input-output relationship, offering limited geometric insight. In contrast, *linear* bound propagation techniques (Wong and Kolter, 2018a;b; Dvijotham et al., 2018; Zhang et al., 2018; Raghunathan et al., 2018; Gehr et al., 2018; Singh et al., 2018; 2019; Wang et al., 2018) which compute bounds via affine relaxations of nonlinear operators offer much tighter enclosures and preserve geometric structure. While more rigorous NN verifiers (Wang et al., 2021b; Zhang et al., 2022; Cheng et al., 2017; Lomuscio and Maganti, 2017; Dutta et al., 2018; Fischetti and Jo, 2017; Tjeng et al., 2019; Xiao et al., 2018; Scheibler et al., 2015) can offer stronger guarantees, their high computational complexity severely limits scalability.

## 3 METHODOLOGY

**Overview.** We introduce the **Guaranteed Inner-and-Outer Meshes (GIOM)** algorithm, which constructs tight, explicit geometric envelopes that bridge the expressive power of neural implicit representations with the efficiency of traditional geometry processors. We also propose **Guaranteed Zero-Level (GIOM-Z)**, a method for approximating the isosurface with theoretical guarantees. Our **key insight** is to integrate scalable NN verification techniques with voxelized spatial hierarchies to efficiently and soundly bound regions of interest. In Section 3.1, we show how bound propagation methods such as CROWN can reformulate geometric queries as NN verification problems. Section 3.2 introduces our bounding shell construction for tight enclosures, and Section 3.3 demonstrates the framework's utility in real-time rendering, physics simulation, and constructive solid geometry.

### 3.1 Geometric Queries as NN Verification Problems.

Geometric queries such as those related to ray-tracing, collision detection, or constructive solid geometry (CSG) operations, can be naturally framed as NN verification problems. Given a 3D SDF $f(x) : \mathbb{R}^3 \to \mathbb{R}$, we have $f(x) < 0$ for points inside the isosurface, $f(x) = 0$ on the surface, and $f(x) > 0$ outside. While an exact SDF also requires that $|f(x)|$ equals the shortest distance from $x$ to the surface, we do not make such assumption in this work. Let $F(x)$ denote the verification property of interest. For example, in collision detection, it is useful to know if a region of space lies entirely outside an object. This can be formulated as:

$$\forall x \in \mathcal{C}, \; F(x) > 0 \quad \text{where } F(x) := f(x) \tag{1}$$

Here, $\mathcal{C} \subset \mathbb{R}^3$ is a region in space (e.g., an axis-aligned bounding box (AABB) defined by corners $x_-$ and $x_+$), and $f(x)$ is the neural SDF evaluated at point $x$. Verifying this property confirms that $\mathcal{C}$ lies strictly outside the object and is thus provably *collision-free*. Conversely, verifying $F(x) < 0$ over $\mathcal{C}$ ensures that the region lies entirely inside the object. If neither condition holds, i.e. $f(x)$ changes sign within the region, then the region must intersect the surface, providing useful cues for contact determination and surface reconstruction. In principle, such properties can be verified by solving global optimization problems that compute $\min_{x \in \mathcal{C}} F(x)$ and $\max_{x \in \mathcal{C}} F(x)$. However, due to the non-convex, nonlinear nature of neural networks, this problem is NP-complete (Katz et al., 2017). Modern NN verification circumvents this by employing scalable techniques like bound propagation, which provide sound interval bounds $f(x) \in [\underline{f}(x), \overline{f}(x)]$ over a region $\mathcal{C}$, enabling global geometric reasoning, such as containment, exclusion, or intersection, with **formal guarantees**.

A particularly effective and representative method of linear bound propagation is CROWN (Zhang et al., 2018), which efficiently computes tight output bounds by backward-propagating affine relaxations through each layer of the network, yielding:

$$\forall x \in \mathcal{C}, \; \underline{\boldsymbol{A}}x + \underline{\boldsymbol{b}} \leq f(x) \leq \overline{\boldsymbol{A}}x + \overline{\boldsymbol{b}} \tag{2}$$

where $\underline{\boldsymbol{A}}, \underline{\boldsymbol{b}}, \overline{\boldsymbol{A}}, \overline{\boldsymbol{b}}$ define lower and upper bounding hyperplanes that can be computed in polynomial time (see Appendix B.1 for details). Optimizing these affine forms over the domain $\mathcal{C} = [x_-, x_+]$ yields tight bounds:

$$y_-^{\mathcal{C}} = \min\{\underline{\boldsymbol{A}}x + \underline{\boldsymbol{b}} \mid x \in \mathcal{C}\}, \; y_+^{\mathcal{C}} = \max\{\overline{\boldsymbol{A}}x + \overline{\boldsymbol{b}} \mid x \in \mathcal{C}\} \tag{3}$$

This provides a sound over-approximation of the true range of $f(x)$ across the entire region, enabling principled, efficient, and certifiable reasoning about geometry from neural SDFs. In the remainder of this work, we show how CROWN facilitates scalable and verifiable solutions to Neural SDF queries.

### 3.2 Shell Extraction via Guaranteed Bounding Meshes

**Motivation.** The goal of *sound* shell extraction is to compute the tightest possible bounding meshes, $\mathcal{M}_-$ and $\mathcal{M}_+$, such that the isosurface $\mathcal{S}$ of a signed distance function (SDF) $f$ lies strictly between them, providing formal guarantees of containment and exclusion that are critical for safety, robustness, and downstream geometric processing. While one might attempt to heuristically "patch" such meshes (e.g., by increasing resolution, adjusting dilation/erosion, or shifting vertices along SDF normals; see Sec. 2), these modifications still cannot provide formal soundness guarantees. In contrast, our bounding meshes are sound by construction. Throughout the rest of this section, we demonstrate that it is possible to extract *sound* inner and outer shells for general neural network architectures. Moreover, we show that this can be achieved *efficiently* using linear bound propagation techniques originally developed for verification tasks as described by (1).

**Voxel Classification and Trimming.** Generating a high-fidelity, explicit volume that tightly encloses or is enclosed within an implicit surface is challenging, but we can start with a basic building block: computing for each *voxel* two polyhedral volumes $\mathcal{V}_-^{\mathcal{C}}$ and $\mathcal{V}_+^{\mathcal{C}}$ such that

$$\mathcal{V}_-^{\mathcal{C}} \subseteq \{x \in \mathcal{C} \mid f(x) \leq 0\} \subseteq \mathcal{V}_+^{\mathcal{C}}, \tag{4}$$

i.e., $\mathcal{V}_-^{\mathcal{C}}$ is strictly inside the implicit surface within $\mathcal{C}$, while $\mathcal{V}_+^{\mathcal{C}}$ is guaranteed to contain it. We obtain these volumes using a simple, two-step procedure that only relies on the linear bounds and over-approximation given in (3).

**(1) Classification.** Use the concretized scalar bounds $y_-^{\mathcal{C}}$ and $y_+^{\mathcal{C}}$ to classify the voxel:

Figure 2: **BaB refinement via voxel trimming.** (a) the bounding meshes $\mathcal{M}_{\pm}^{C_0}$ (blue and green) of an SDF, the boundary of which cuts through an UNK voxel $\mathcal{C}$, experiences BaB refinement. (b) UNK subdomains like $\mathcal{C}_2$ are trimmed by bounding meshes. (c) The union of UNK voxels produce tighter bounding volumes $\mathcal{V}_{\pm}^{C_0'}$ and meshes $\mathcal{M}_{\pm}^{C_0'}$ than before BaB and voxel trimming.

- If $y_-^{\mathcal{C}} > 0$, then $f(x) > 0$ for all $x \in \mathcal{C}$. Mark $\mathcal{C}$ as POS and set $\mathcal{V}_-^{\mathcal{C}} = \mathcal{V}_+^{\mathcal{C}} = \emptyset$.
- If $y_+^{\mathcal{C}} < 0$, then $f(x) < 0$ for all $x \in \mathcal{C}$. Mark $\mathcal{C}$ as NEG and set $\mathcal{V}_-^{\mathcal{C}} = \mathcal{V}_+^{\mathcal{C}} = \mathcal{C}$.
- Otherwise mark $\mathcal{C}$ as UNK (potentially intersecting).

**(2) Voxel trimming for UNK voxels.** For an UNK voxel we can fall back to the trivial choice $\mathcal{V}_+^{\mathcal{C}} = \mathcal{C}$, $\mathcal{V}_-^{\mathcal{C}} = \emptyset$, but this is often overly coarse. Instead, we propose to reuse the affine linear bounds to trim $\mathcal{C}$ into tighter polyhedral enclosures:

$$\mathcal{V}_+^{\mathcal{C}} := \{x \in \mathcal{C} \mid \underline{A}x + \underline{b} \le 0\}, \ \mathcal{V}_-^{\mathcal{C}} := \{x \in \mathcal{C} \mid \overline{A}x + \overline{b} \le 0\} \tag{5}$$

By construction these sets are conservative: $\mathcal{V}_-^{\mathcal{C}}$ is guaranteed to lie inside the true negative region and $\mathcal{V}_+^{\mathcal{C}}$ to contain the surface portion inside $\mathcal{C}$. The trimmed volume $\mathcal{V}_+^{\mathcal{C}} \setminus \mathcal{V}_-^{\mathcal{C}}$ is a convex polytope that conservatively contains any isosurface segment crossing $\mathcal{C}$.

**Branch-and-Bound for Voxel Verification.** Given a collection of bounding volumes, $\mathcal{B} = \{\mathcal{B}_0 \ldots \mathcal{B}_n\}$, we define the inner and outer bounding volumes of the true implicit volume, $\mathcal{V}$, as:

$$\mathcal{V}_- = \bigcup_i \mathcal{V}_-^{\mathcal{B}_i}, \qquad \mathcal{V}_+ = \bigcup_i \mathcal{V}_+^{\mathcal{B}_i} \tag{6}$$

To construct these approximations efficiently, we apply a branch-and-bound (BaB) process starting from an initial Axis-Aligned Bounding Box (AABB) (See Fig. 2. Only UNK voxels can intersect the implicit surface, therefore only this class of voxels are recursively subdivided along their largest dimension. To improve scalability, we introduce an early termination criterion: terminate further subdivision if (1) both bounding planes intersect the voxel, and (2) the distance between their intersections with the voxel is below a fixed threshold (Alg. 1). Once BaB terminates, we union all POS, NEG, and early-stopped UNK voxels to construct the final inner and outer bounding volume as described by (6). In practice, this corresponds to merging all polyhedral meshes derived from voxel trimming, yielding the *bounding surface meshes* $\mathcal{M}_- := \partial \mathcal{V}_-$

---

**Algorithm 1** Adaptive Split

**Input**: neural implicit surface $f$, AABB with range $[l, u]$, max split depth $D$.
**Output**: bounding volumes $\mathcal{V}_-, \mathcal{V}_+$
$\mathcal{V}_- \leftarrow \emptyset, \mathcal{V}_+ \leftarrow \emptyset, \mathcal{B} \leftarrow \{[l, u]\}, d \leftarrow 0$
**while** $d < D$ **do**
    **for** $\mathcal{B}_i$ in $\mathcal{B}$ **do**
        $\mathcal{B} \leftarrow \mathcal{B} \setminus \{\mathcal{B}_i\}$
        $T, \overline{f}, \underline{f} \leftarrow \text{ComputeBounds}(x_-, x_+)$
        **if** $T = $ POS **then continue**
        **else if** $T = $ NEG **then** $\mathcal{V}_- \leftarrow \mathcal{V}_- \cup \mathcal{B}_i$
        **else if** $\text{EarlyStop}(\mathcal{B}_i, \overline{f}, \underline{f}) \lor d = D - 1$ **then**
            $\mathcal{V}_-^{\mathcal{B}_i}, \mathcal{V}_+^{\mathcal{B}_i} \leftarrow \text{Trim}(\mathcal{B}_i, \overline{f}, \underline{f})$
            $\mathcal{V}_- \leftarrow \mathcal{V}_- \cup \mathcal{V}_-^{\mathcal{B}_i}, \mathcal{V}_+ \leftarrow \mathcal{V}_+ \cup \mathcal{V}_+^{\mathcal{B}_i}$
        **else**
            $\mathcal{B} \leftarrow \mathcal{B} \cup \text{MaxDimSplit}(\mathcal{B}_i)$
    $d \leftarrow d + 1$
**return** $\mathcal{V}_-, \mathcal{V}_+$

---

and $\mathcal{M}_+ := \partial \mathcal{V}_+$. This final union can be performed efficiently with existing mesh processing libraries such as Trimesh (et al., 2019).

**Guaranteed Zero-Level Extraction via GIOM.** We also introduce *Guaranteed Zero-Level* (GIOM-Z), a method that utilizes the linear bounds to approximate the isosurface, with tight precision guarantees. For each UNK voxel, we define an approximate linear surface by averaging the

upper and lower bounds: $\boldsymbol{A}_0 = (\underline{\boldsymbol{A}} + \overline{\boldsymbol{A}})/2$, $b_0 = (\underline{b} + \overline{b})/2$, yielding an approximation of the true implicit surface within the voxel $\mathcal{C}$ as $\mathcal{V}_0^{\mathcal{C}} := \{x \in \mathcal{C} \mid \boldsymbol{A}_0 x + b_0 \le 0\}$. For POS and NEG voxels, we set $\mathcal{V}_0^{\mathcal{C}} = \emptyset$ and $\mathcal{C}$, respectively. We extract a surface mesh by taking the boundary of the implicit volume, i.e. $\mathcal{M}_0 := \partial \mathcal{V}_0$, where $\mathcal{V}_0 := \cup_i \mathcal{V}_0^{\mathcal{B}_i}$. Unlike marching cubes, whose precision guarantee are the size of the smallest voxel, our GIOM-Z has a theoretical guarantee *equal to the smallest distance between bounding geometries*.

## 3.3 GIOM for Geometry Queries

With GIOM, we can accelerate various geometry queries on neural SDFs or implicit surfaces, and the key insight is to use the guaranteed bounding volumes $\mathcal{V}_-$ and $\mathcal{V}_+$ for a complex neural surface.

**GIOM-Accelerated Rendering.** The ray casting problem on an SDF $f$ can be defined as finding the smallest $t$ given ray root $p \in \mathbb{R}^3$ and ray direction $r \in \mathbb{R}^3$ such that $f(p + tr) = 0$. Given a pair of inner and outer shells, we present a fast Algorithm 2 to identify the intersection point $p' = p + t'r$ between a ray and an implicit surface encoded by an SDF with precision guarantee $\delta$ (e.g. $f(p + (t' + \delta)r) < 0 < f(p + t'r)$). We first extract **inner and outer bounding meshes** $\mathcal{M}_-$ and $\mathcal{M}_+$ from bounding volumes $\mathcal{V}_-$ and $\mathcal{V}_+$. For each ray $(p, r)$ that has at least two intersections $p_i$ and $p_o$ with $\mathcal{M}_- \cup \mathcal{M}_+$, we identify it as a candidate ray that might hit the true surface. By sampling with an interval of $\delta$ inside the range $(p_i, p_o)$ and querying $f$ with the samples, we are *guaranteed* to find the intersection between the ray and the true surface with precision of $\delta$. If the intersection does not occur in the current range, we keep the status of the ray undetermined and proceed to the next iteration. Since our bounding shells are tight, we can afford to query all the samples along every ray in each iteration in parallel, even on a commodity GPU.

**Algorithm 2** Efficient Ray Casting with GIOM

Input: neural implicit surface $f$, inner shell mesh $\mathcal{M}_-$, outer shell mesh $\mathcal{M}_+$, ray root $p$, ray direction $r$, and precision $\delta$.
**while** True **do**
    $p_i \leftarrow (\mathcal{M}_- \cup \mathcal{M}_+).\text{intersect}(p, r)$
    $p_o \leftarrow (\mathcal{M}_- \cup \mathcal{M}_+).\text{intersect}(p_i, r)$
    **if** $p_o$ is None **then**
        // The ray exits the bounding shells.
        **return** $p_i$
    $P \leftarrow \{p_i, p_i + \delta r \dots p_o\}$
    $D \leftarrow f(P)$
    **for** $j$ in $0 \dots |D| - 2$ **do**
        **if** $d_{j+1} < 0 < d_j$ **then**
            // The ray intersects the isosurface.
            **return** $p_j$
    // The ray was grazing.
    // Update ray root.
    $p \leftarrow p_o$

Figure 3: **GIOM bounds accelerate implicit ray casting.** Top: naive neural sphere tracing. Bottom: GIOM shells tightly bound the surface's zero crossing. Right: skipping ray interval with GIOM shells cuts # of network queries significantly ($3 \to 3$, $6 \to 3$, $8 \to 0$).

Fig. 3 depicts how GIOM bounding shell can accelerate the required (#) of samples when rendering a neural implicit surface. The decrease in sample counts is positively correlated with the tightness of the shell, as visualized in Fig. 4. By casting rays directly on GIOM-Z, we can further boost efficiency at a subtle cost of quality.

**GIOM-Accelerated Collision Detection.** While neural SDFs can be baked into a voxel database for real-time physics simulation, collision detection efficiency remains a bottleneck, especially in large-scale particle collision and mesh-SDF collision. To address this issue, we utilize our outer bounding mesh as a spatial hierarchy, ensuring that an object not colliding with the mesh will never collide with the object defined by the neural SDF. Since checking collisions with a mesh can be done

very efficiently, and our bound mesh is also much tighter compared to loose approximations such as AABB boxes, GIOM can significantly improve collision detection efficiency.

Table 1: CSG bounding volume computation and conventional formulation.

| CSG Type | $f_1 \cup f_2$ | $f_1 \cap f_2$ | $f_1 - f_2$ |
|---|---|---|---|
| Bounding Volumes | $\mathcal{V}_- = \mathcal{V}_-^{(1)} \cup \mathcal{V}_-^{(2)}$ $\mathcal{V}_+ = \mathcal{V}_+^{(1)} \cup \mathcal{V}_+^{(1)}$ | $\mathcal{V}_- = \mathcal{V}_-^{(1)} \cap \mathcal{V}_-^{(2)}$ $\mathcal{V}_+ = \mathcal{V}_+^{(1)} \cap \mathcal{V}_+^{(2)}$ | $\mathcal{V}_- = \mathcal{V}_-^{(1)} \setminus \mathcal{V}_+^{(2)}$ $\mathcal{V}_+ = \mathcal{V}_+^{(1)} \setminus \mathcal{V}_-^{(2)}$ |
| Conventional Formulae | $f = \min(f_1, f_2)$ | $f = \max(f_1, f_2)$ | $f = \max(f_1, -f_2)$ |

**GIOM-Accelerated CSG Operation.** CSG (constructive solid geometry) requires accurate computation of the union, intersection, or difference of two SDFs $f_1$ and $f_2$, in the form of one new implicit surface $f$. The conventional definition of the new implicit surface $f$ from $f_1$ and $f_2$ (last row of Tab. 1) yields correct isosurfaces but distorts the distance field, making the SDF unsuitable for rendering or physics simulation. To address this, CSG-nSDF (Marschner et al., 2023a) trains a neural network with geometric constraints to encode an exact neural SDF of CSG, but this optimization-based approach cannot perfectly align with the ground truth (Fig. 8).

We propose an optimization-free algorithm (Alg. 3) to approximate the exact signed distance field resulting from a CSG operation between two SDFs. Given two SDFs $f_1$ and $f_2$, and their bounding volumes, $\mathcal{V}_\pm^{(1)}$ and $\mathcal{V}_\pm^{(2)}$, we can easily compute the new bounding volumes $\mathcal{V}_\pm$ (Tab. 1). After we extract the bounding meshes $\mathcal{M}_-$ and $\mathcal{M}_+$, we are ready to query for signed distance values. For points that lie between the bounding meshes $\mathcal{M}_-$ and $\mathcal{M}+$, we use the conventional formulae as near-surface approximation. For points outside this region, we start by finding the closest point on the bounding meshes. We then compute the total distance as the sum of two components: the distance from the query point to this closest mesh point, and the distance from the mesh point to the isosurface. Finally, we assign a sign to the computed distance based on whether the query point is inside or outside the surface.

---

**Algorithm 3** CSG with GIOM

**Input:** query point $x$, SDF $f_1$ and $f_2$, bounding volumes $\mathcal{V}_\pm^{(1)}$ and $\mathcal{V}_\pm^{(2)}$, and boolean operator $\circ$.
**Output:** signed distance from $x$ to CSG surface
$\mathcal{V}_\pm, f \leftarrow f_1 \circ f_2$ per Table 1
$\mathcal{M}_- \leftarrow \partial \mathcal{V}_-, \mathcal{M}_+ \leftarrow \partial \mathcal{V}_+$
**if** $x \in \mathcal{V}_+ \setminus \mathcal{V}_-$ **then**
    **return** $f(x)$
**else**
    $x_d \leftarrow \text{ClosestPoint}(x, \mathcal{M}_- \cup \mathcal{M}_+)$
    **return** $f(x_d) + \text{sign}(x \notin \mathcal{V}_+) \cdot \|x - x_d\|$

---

# 4 EXPERIMENTS

We validate our proposed GIOM method across three distinct neural implicit geometry-based visual computing tasks: neural rendering, physical simulation, and constructive solid geometry (CSG); we compare its performance and efficiency against existing methods in each task. Additional visualizations and video results are provided in the supplementary material.

**Real-Time Rendering** Given a neural SDF and camera settings (position, direction, and front-of-view (FOV) angle), our goal is to render high-resolution, high-fidelity images corresponding to the input geometry. We report SSIM, PSNR, and RMSE as image quality metrics, along with FPS to evaluate rendering efficiency. We compare against four baselines: sphere tracing (ST) (Hart, 1996), affine-arithmetic-based interval tracing (IT) (Sharp and Jacobson, 2022), adaptive shells for SDFs, and 0-level set approximation via marching cubes. Adaptive shells are originally used for accelerating queries on NeRF, but we use them here to denote inner and outer mesh extraction via dilation and erosion with marching cubes (the adaptation to our setting is described in Appendix C.2). We manually tune the dilation and erosion extents until the bounding meshes no longer intersect the surface mesh visually and combine them with Alg. 2 as the third baseline.

Fig. 10 shows that (1) GIOM outperforms every baseline in time while being comparable in quality, and that (2) GIOM-Z excels 0 level MC in quality with the same or better efficiency. Tab 2 consolidates the observation with overall trends. Although adaptive shells can achieve slightly better PSNR than GIOM, the latter beats it in efficiency by $3\times$. GIOM-Z beats 0-level MC in all magnitudes due

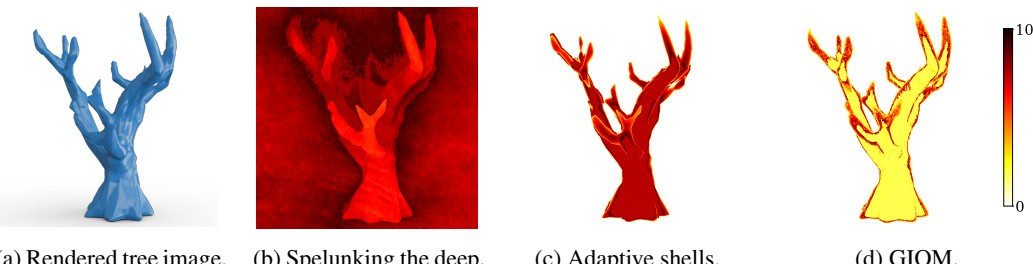

(a) Rendered tree image.  (b) Spelunking the deep.  (c) Adaptive shells.  (d) GIOM.

Figure 4: **Comparison of sample counts per pixel between high-quality exact rendering with adaptive shells and GIOM.** Compared to adaptive shells, which obtain bounds via a dilation-and-erosion procedure, GIOM is constructed using a neural-network bounding algorithm, resulting in tighter bounds and thus reducing the number (#) of samples per ray.

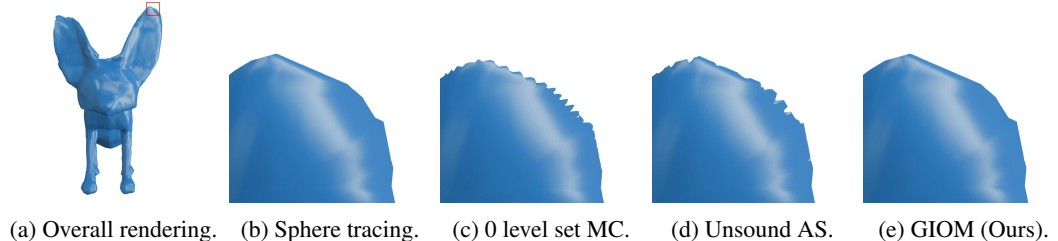

(a) Overall rendering.  (b) Sphere tracing.  (c) 0 level set MC.  (d) Unsound AS.  (e) GIOM (Ours).

Figure 5: **Qualitative comparison of rendering details.** Compared to zero-set Marching Cube (c) and potentially unsound bounding meshes produced by Adaptive Shell (d), GIOS-based ray casting (Alg. 2) preserves thin-surface details and achieves performance comparable to the exact sphere tracing (b) using sphere tracing of Neural SDFs, while being 70 times faster.

to the adaptive refinement enabled by tight bounding planes. Fig. 5 demonstrates the advantage of tight bounding meshes and guaranteed ray casting algorithm in rendering fine details. The tip of the ear of the fox is almost volumeless, and marching cubes leads to missing volumes. Among all the guaranteed ray casting algorithms, GIOM remains the fastest.

**Real-Time Physics Simulation**     We run discrete particle, cloth, and continuous mesh-SDF collision simulations to demonstrate two dimensions of the benefits of GIOM in physics simulation: efficiency and quality. We measure the time spent on physics simulation and visualize the simulated collision. For particle collision, we compare our GIOM against two baselines: voxel SDF (sampled from neural SDF) without any acceleration and with Axis-Aligned Bounding Box (AABB). For cloth simulation, we compare GIOM with Euler integrator against with only Euler or XPBD integrator. For mesh-SDF continuous collision detection (CCD), we compare against a state-of-the-art method (Pelletier-Guénette et al., 2025).

Tab. 3 suggests that the overall runtime of collision detection decreases and stabilizes with our GIOM. We achieve a significant efficiency boost on the tree object. Because the tree has the most complex structure of all four objects, the tightness of GIOM becomes apparently advantageous once the particles come into contact with the query body (Fig. 6c). Fig. 6a further validates the effectiveness of GIOM on the tree object. In addition to the particle simulation, we run a cloth simulation with GIOM in Fig. 7c. GIOM is efficient and tunneling-free; in contrast, directly simulating with the voxel SDF causes unwanted intersections (Fig. 7a, Fig. 7b). We also show that CCD time can be reduced by replacing expensive neural SDF queries with cheap tests against GIOM (Tab. 4).

**Constructive Solid Geometry**     Given two SDFs, our goal is to reconstruct the signed distance field of their CSG. The signed distance field should not only model the new border correctly but also be exact everywhere in the space. We measure reconstruction time and error, as well as the time to query the reconstructed SDF. We compare our method with CSG-nSDF (Marschner et al., 2023a). We use the open-source model on the union of a square and a circle provided in the official repository (Marschner et al., 2023b).

Table 2: Quantitative comparisons of implicit rendering quality and speed.

| Method | Type | RMSE↓ | PSNR↑ | SSIM↑ | FPS↑ |
|---|---|---|---|---|---|
| ST (Groundtruth) | Ray Marching | 0.00 | $\infty$ | 1.000 | $0.21 \pm 0.12$ |
| IT (Spelunking) | Ray Marching | **0.01** | $46.51 \pm 5.68$ | $\mathbf{0.999 \pm 0.001}$ | $0.24 \pm 0.16$ |
| AdaptiveShells | Shell Accel. | 0.01 | $\mathbf{46.00 \pm 5.20}$ | $0.998 \pm 0.001$ | $6.0 \pm 5.6$ |
| GIOM (Ours) | Shell Accel. | 0.01 | $45.87 \pm 5.15$ | $0.998 \pm 0.001$ | $\mathbf{18 \pm 10}$ |
| 0 Level MC | Mesh Approx. | 0.03 | $32.14 \pm 3.42$ | $0.985 \pm 0.012$ | $181 \pm 113$ |
| GIOM-Z (Ours) | Mesh Approx. | **0.01** | $\mathbf{37.80 \pm 3.34}$ | $\mathbf{0.995 \pm 0.004}$ | $\mathbf{202 \pm 69}$ |

Table 3: Particle collision detection time (ms) for different methods and objects.

| Method | fox | cat | tree | koala | Mean |
|---|---|---|---|---|---|
| SDF only | $6.00 \pm 1.36$ | $5.94 \pm 1.43$ | $15.10 \pm 5.87$ | $5.94 \pm 1.68$ | $8.24 \pm 8.54$ |
| AABB | $3.52 \pm 1.96$ | $2.69 \pm 1.53$ | $12.09 \pm 6.29$ | $3.42 \pm 2.03$ | $5.43 \pm 8.49$ |
| Ours | $\mathbf{2.00 \pm 1.02}$ | $\mathbf{2.13 \pm 1.15}$ | $\mathbf{3.09 \pm 2.41}$ | $\mathbf{1.95 \pm 1.16}$ | $\mathbf{2.29 \pm 1.80}$ |

Table 4: Quantitative comparisons of CCD efficiency and bandwidth.

| Method | Stepping time (ms) per frame↓ | | | | # Tri tests↓ | |
|---|---|---|---|---|---|---|
| | min | median | max | total | Mesh | nSDF |
| CCD | 52.89 | 2940.50 | 8504.33 | $5.34 \times 10^5$ | 0 | 9265 |
| Ours | **13.69** | **464.44** | **4366.14** | $\mathbf{1.90 \times 10^5}$ | 9710 | **1695** |

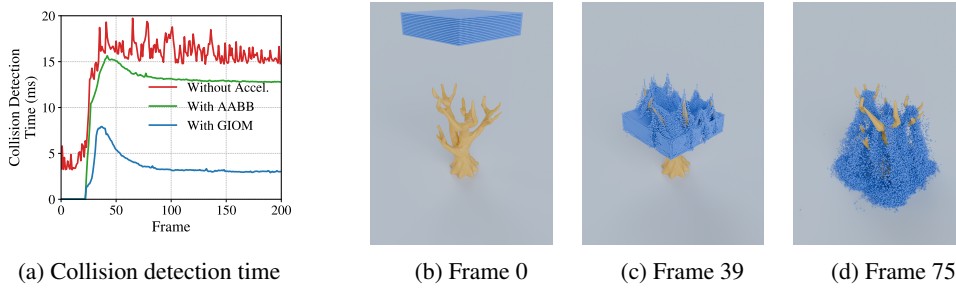

(a) Collision detection time      (b) Frame 0      (c) Frame 39      (d) Frame 75

Figure 6: **Accelerated physical simulation on a complex neural implicit surface (tree) with GIOM.** Top: Runtime per frame during the episode. GIOM is significantly faster than the conventional AABB spatial-hierarchy bound, which provides negligible speedup compared to directly querying voxel SDFs. This difference becomes particularly pronounced when more particles collide with the body (see Fig. 6c, 6d). Bottom: Qualitative results. Thanks to the guaranteed bound, the simulation exhibits realistic behavior with no noticeable differences compared to exact SDF queries.

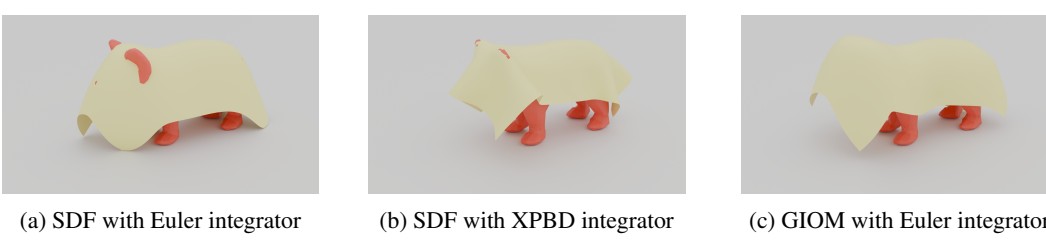

(a) SDF with Euler integrator      (b) SDF with XPBD integrator      (c) GIOM with Euler integrator

Figure 7: **Qualitative results of cloth simulation.** Compared to voxel-SDF-based approaches, the combination of GIOM and Euler integrator yields a more visually satisfying result.

Fig. 8d shows that we outperform CSG-nSDF in reconstruction efficiency and accuracy. The one minute reconstruction time includes the overhead of bounding polygon computation. Fig. 8 offers some clues for the drastic different in reconstruction accuracy. CSG-nSDF has visible errors, while our method has not visual difference from the ground truth.

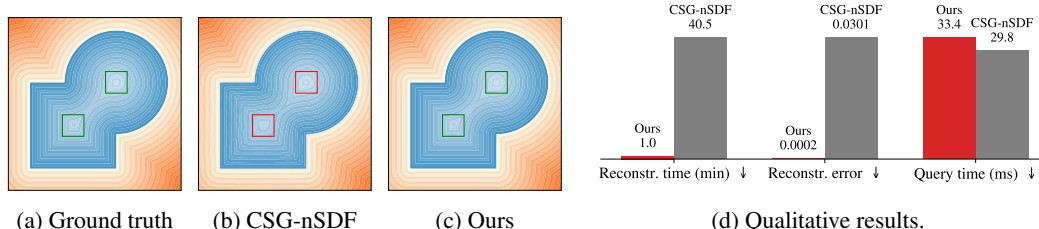

(a) Ground truth    (b) CSG-nSDF    (c) Ours    (d) Qualitative results.

Figure 8: **Qualitative and quantitative results for constructive solid geometry.** Left (a–c): Compared to baseline (b), GIOM produces a more accurate signed distance field in the central regions. Right: Since no neural SDF retraining is needed, our reconstruction time is significantly lower while being more accurate. The query time is only marginally higher than baseline.

## 5 CONCLUSION & LIMITATIONS

In this paper, we proposed Guaranteed Inner-and-Outer Meshes (GIOM), a framework for generating tight, sound, and explicit bounding volumes of neural implicit surfaces. We leverage GIOM to accelerate a variety of geometric queries, including ray casting, collision detection, and CSG operations. GIOM offers a favorable balance between efficiency and quality across these tasks.

**Limitations.** GIOM is currently limited to static neural implicits. Extending it to support time-varying fields would be a promising direction for future work, potentially enabling applications such as dynamic collision detection, animation rendering, and swept volume estimation.

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

# A LLM USAGE

The roles of LLM in this work are: simple baseline code generation, result post-processing, and language revision. LLM is not involved in the development of core methodology, design of evaluation, analysis of results, or any other significant component.

# B FORMULATIONS

## B.1 NN VERIFICATION BACKGROUND

In this section, we provide a more detailed prescription describing how a linear bound propagation method such as CROWN is able to provide sound, linear bounds for a ReLU feed-forward neural network. These ideas may be generalized to more complex neural network architectures with more diverse activation functions, and we provide the appropriate references to works that are well-suited to describing these extensions.

**The MIP Formulation** The mixed integer programming (MIP) formulation is the root of many NN verification algorithms. Given the ReLU activation function's piecewise linearity, the model requires binary encoding variables, or ReLU indicators, $s$, only for unstable neurons. We formulate the optimization problem aiming to minimize the function $f(\boldsymbol{x})$, subject to a set of constraints that encapsulate the DNN's architecture and the perturbation limits around a given input $\boldsymbol{x}$, as follows:

$$f^\star = \min_{\boldsymbol{z}, \hat{\boldsymbol{z}}, \boldsymbol{s}} f(\boldsymbol{x}) \qquad \text{s.t. } f(\boldsymbol{x}) = \boldsymbol{z}^{(L)}; \hat{\boldsymbol{z}}^{(0)} = \boldsymbol{x} \in \mathcal{C} \tag{7a}$$

$$\hat{\mathbf{z}}^{(i)} = \mathbf{W}^{(i)}\hat{z}^{(i-1)} + \mathbf{b}^{(i)}; \quad i \in [L] \tag{7b}$$

$$\mathcal{I}^{+(i)} := \{j \ : \ \boldsymbol{l} \geq 0\} \tag{7c}$$

$$\mathcal{I}^{-(i)} := \{j \ : \ \boldsymbol{u} \leq 0\} \tag{7d}$$

$$\mathcal{I}^{(i)} := \{j \ : \ \boldsymbol{l} < 0, \boldsymbol{u} > 0\} \tag{7e}$$

$$\mathcal{I}^{+(i)} \cup \mathcal{I}^{-(i)} \cup \mathcal{I}^{(i)} = \mathcal{J}^i \tag{7f}$$

$$\hat{\boldsymbol{z}} \geq 0; j \in \mathcal{I}^{(i)}, i \in [L-1] \tag{7g}$$

$$\hat{\boldsymbol{z}} \geq \boldsymbol{z}; j \in \mathcal{I}^{(i)}, i \in [L-1] \tag{7h}$$

$$\hat{\boldsymbol{z}} \leq \boldsymbol{u}\boldsymbol{s}; j \in \mathcal{I}^{(i)}, i \in [L-1] \tag{7i}$$

$$\hat{\boldsymbol{z}} \leq \boldsymbol{z} - \boldsymbol{l}(1 - \boldsymbol{s}); j \in \mathcal{I}^{(i)}, i \in [L-1] \tag{7j}$$

$$\boldsymbol{s} \in \{0, 1\}; j \in \mathcal{I}^{(i)}, i \in [L-1] \tag{7k}$$

$$\hat{\boldsymbol{z}} = \boldsymbol{z}; j \in \mathcal{I}^{+(i)}, i \in [L-1] \tag{7l}$$

$$\hat{\boldsymbol{z}} = 0; j \in \mathcal{I}^{-(i)}, i \in [L-1] \tag{7m}$$

To initialize intermediate bounds for each neuron, we replace the original objective $f(\boldsymbol{x})$ with the neuron's pre-activation value $\boldsymbol{z}$. This lets us solve the following bounds for every neuron $j$ in layer $i$, with $i \in [L-1]$ and $j \in \mathcal{J}^{(i)}$:

$$\boldsymbol{l} = \min_{\boldsymbol{x} \in \mathcal{C}} f(\boldsymbol{x}), \quad \boldsymbol{u} = \max_{\boldsymbol{x} \in \mathcal{C}} f(\boldsymbol{x}). \tag{8}$$

Here, the set $\mathcal{J}^{(i)}$ comprises all neurons in layer $i$, which can be categorized into three groups: 'active' ($\mathcal{I}^{+(i)}$), 'inactive' ($\mathcal{I}^{-(i)}$), and 'unstable' ($\mathcal{I}^{(i)}$).

Next, the MIP formulation is initialized with the constraints

$$\boldsymbol{l} \ \leq \ \boldsymbol{z} \ \leq \ \boldsymbol{u} \tag{9}$$

across all neurons and layers $i$. These bounds can be computed recursively, propagating from the first layer up to the $i$-th layer. However, since MIP problems involve integer variables, they are generally NP-hard, reflecting the computational challenge of this approach.

**The LP and Planet relaxation.** By relaxing the binary variables in equation 7k to $s \in [0,1], j \in \mathcal{I}^{(i)}, i \in [L-1]$, we can get the LP relaxation formulation. By replacing the constraints in equation 7i, equation 7j, equation 7k with

$$\hat{x} \leq \frac{u}{u-l}(x-l); \ j \in \mathcal{I}^{(i)}, i \in [L-1], \tag{10}$$

we can eliminate the $s$ variables and get the well-known Planet relaxation formulation. Both of these two relaxations are solvable in polynomial time to yield lower bounds.

**Linear Bound Propagation**. We now consider linear bound propagation methods which bound a NN in a recursive fashion such as CROWN. For a feed-forward network, CROWN will sequentially provide a bound on all vectors $z^{(i)}, i \in [L]$, by back-propagating linear relationships from the $i^{th}$ layer back to input $x$. These bounds are described as:

$$\underline{z}^{(i)} := \min_{x \in \mathcal{C}} \underline{A}^{(i)} x + \underline{c}^{(i)} \leq z^{(i)}, \quad \overline{z}^{(i)} := \max_{x \in \mathcal{C}} \overline{A}^{(i)} x + \overline{c}^{(i)} \geq z^{(i)} \tag{11}$$

When $\mathcal{C}$ is an $\ell_\infty$ box, we may "concretize" the lower and upper bounds using Hölder's inequality: $\underline{z}^{(i)} = \underline{A}^{(i)} \hat{x} - |\underline{A}^{(i)}| \epsilon + \underline{c}^{(i)}$ and $\overline{z}^{(i)} = \overline{A}^{(i)} \hat{x} + |\overline{A}^{(i)}| \epsilon + \overline{c}^{(i)}, i \in [L]$, where $\underline{A}^{(i)} \in \mathbb{R}^{n_i \times n_0}$ and $\underline{c}^{(i)} \in \mathbb{R}^{n_i}$. Once concretized, the post-activation neuron, $\hat{z}$, at intermediate layers may be bounded using the Planet relaxation as described in equation equation 10. Bound propagation is not limited to feed-forward networks, and readers are deferred to the LiRPA framework in Xu et al. (2020a) which describes how bound propagation algorithms may be applied to more general networks.

In a feedforward network, $\underline{A}^{(i)}, \overline{A}^{(i)}, \underline{c}^{(i)}$ and $\overline{c}^{(i)}$ must be derived for every linear layer preceding an activation layer, as well as the final layer of the network. In order to derive the hyperplane coefficients ($\underline{A}^{(i)}/\overline{A}^{(i)}$) and biases ($\underline{c}^{(i)}/\overline{c}^{(i)}$), at this $i^{th}$ layer, all preceding activation layers must have already had their inputs bounded. The following lemma describes how a ReLU activation layer may be relaxed which will be useful for defining bounding hyperplanes, $\underline{A}^{(i)}, \overline{A}^{(i)}, \underline{c}^{(i)}$ and $\overline{c}^{(i)}$ .

(Relaxation of a ReLU layer in CROWN). Given the lower and upper bounds of $z_j^{(i-1)}$, denoted as $l_j^{(i-1)}$ and $u_j^{(i-1)}$, respectively, the linear layer proceeding the ReLU activation layer may be lower-bounded element-wise by the following inequality:

$$z^{(i)} = W^{(i)} \sigma(z^{(i-1)}) \geq W^{(i)} D^{(i-1)} z^{(i-1)} + W^{(i)} \underline{b}^{(i-1)} \tag{12}$$

where $D^{(i-1)}$ is a diagonal matrix with shape $\mathbb{R}^{n_{i-1} \times n_{i-1}}$ whose off-diagonal entries are 0, and on-diagonal entries are defined as:

$$D_{j,j}^{(i-1)} := \begin{cases} 1, & l_j^{(i-1)} \geq 0 \\ 0, & u_j^{(i-1)} \leq 0 \\ \alpha_j^{(i-1)}, & l_j^{(i-1)} < 0 < u_j^{(i-1)} \text{ and } W \geq 0 \\ \frac{u_j^{(i-1)}}{u_j^{(i-1)} - l_j^{(i-1)}}, & l_j^{(i-1)} < 0 < u_j^{(i-1)} \text{ and } W < 0 \end{cases} \tag{13}$$

and $\underline{b}_j^{(i-1)}$ is a vector with shape $\mathbb{R}^{n_{i-1}}$ whose elements are defined as:

$$\underline{b}_j^{(i-1)} := \begin{cases} 0, & l_j^{(i-1)} > 0 \text{ or } u_j^{(i-1)} \leq 0 \\ 0, & l_j^{(i-1)} < 0 < u_j^{(i-1)} \text{ and } W \geq 0 \\ -\frac{u_j^{(i-1)} l_j^{(i-1)}}{u_j^{(i-1)} - l_j^{(i-1)}}, & l_j^{(i-1)} < 0 < u_j^{(i-1)} \text{ and } W < 0 \end{cases} \tag{14}$$

In the above definitions, $\alpha_j^{(i-1)}$ is a parameter in range $[0,1]$ and may be fixed or optimized as in Xu et al. (2020b).

For the $j^{th}$ ReLU at the $(i-1)^{th}$ layer, it's result may be bounded as follows:

$$\alpha_j^{(i-1)} z_j^{(i-1)} \leq \sigma(z_j^{(i-1)}) \leq \frac{u_j^{(i-1)}}{u_j^{(i-1)} - l_j^{(i-1)}}(z_j^{(i-1)} - l_j^{(i-1)}). \tag{15}$$

The right-hand side holds as this is the Planet-relaxation. For the left-hand side, we first consider when $z_j^{(i-1)} \leq 0$. For every input in this range, the result of the ReLU is $\sigma(z_j^{(i-1)}) = 0$. $\alpha_j^{(i-1)} z$ forms a line for which inputs in this range will always produce a non-positive result when $\alpha_j^{(i-1)} \in [0, 1]$. For inputs in the range $z_j^{(i-1)} \geq 0$, the result of the ReLU is $\sigma(z_j^{(i-1)}) = z_j^{(i-1)}$. This result is never exceeded by $\alpha_j^{(i-1)} z_j^{(i-1)}$ when $\alpha_j^{(i-1)} \in [0, 1]$.

When the result, $\sigma(z_j^{(i-1)})$, is multiplied by a scalar such as $W$, a valid lower-bound of $W\sigma(z_j^{(i-1)})$ requires a lower bound on $\sigma(z_j^{(i-1)})$ when $W \geq 0$, and an upper bound on $\sigma(z_j^{(i-1)})$ when $W < 0$. Such lower and upper bounds are indeed produced by $D_{j,j}^{(i-1)}$ and $\underline{b}_j^{(i-1)}$, whose definitions are derived from the inequality displayed in equation equation 15. This concludes the proof.

Lemma B.1 suggests a recursive approach to bounding a neural network as the bounds at the $i^{\text{th}}$ layer depends on the bounds of the layer preceding it due to the dependence on $l_j^{(i-1)}$ and $u_j^{(i-1)}$. This is indeed the case, and we may define our hyperplane coefficients as $\underline{A}^{(i)} = \Omega^{(i,1)} W^{(1)}$ where

$$\Omega^{(i,k)} := \begin{cases} W^{(i)} D^{(i-1)} \Omega^{(i-1)}, & i > k \\ I, & i = k \end{cases} \tag{16}$$

To collect the remaining terms, we set $\underline{c}^{(i)} = \sum_{k=2}^{i} \left( \Omega^{(i,k)} W^{(k)} \underline{b}^{(k-1)} \right) + \sum_{k=1}^{i} \left( \Omega^{(i,k)} b^{(k)} \right)$. To obtain an upper bound, Lemma B.1 and its proof may be adjusted accordingly where appearances of the inequalities $W^{(i)} \geq 0$ and $W^{(i)} < 0$ are flipped. In doing so, we may repeat this recursive process in order to obtain $\overline{A}^{(i)}$ and $\overline{c}^{(i)}$.

Though we have described how a ReLU feedforward network may be bounded, appropriately updating the definitions of $D^{(i)}$ and $\underline{b}^{(i)}$ allows feedforward networks with general activation functions (that act element-wise) to be bounded. Such a general formulation is described in Zhang et al. (2018) that is similar to the template described above, and goes into further detail on how this formulation may be extended to *quadratic* bound propagation.

## C  IMPLEMENTATIONS

### C.1  MESH METADATA

Bound computation is achieved via auto_LiRPA, and geometry processing, including voxel trimming and union, is completed via Trimesh et al. (2019). The number of vertices and faces are kept at the same level for fairness (see Appendix). We have four objects encoded with neural SDF: fox, cat, koala, and tree. The fox SDF is a pre-trained one from the codebase of Sharp and Jacobson (2022), and the other three were trained with the training script provided by Sharp and Jacobson (2022), using training data from Stein (2024). In Tables 5, 7, 9, and 17, we present the max mesh resolution (Res.), total bounding mesh construction time in seconds (Time), bounding mesh type (I/O for inner/outer), number of mesh vertices (Vertices), number of mesh faces (Faces), memory cost in MB (Mem.), maximum and minimum signed distance (Max SD and Min SD) from mesh surface samples to the bounded surface, and mean unsigned distance (Mean D) from mesh surface samples to the bounded surface. In addition, for AdaptiveShells, we present the aforementioned metrics with the minimum dilation/erosion extent (DE) required to achieve empirical robustness in Tables 6, 8, 10, and 18. The minimum robust dilation and erosion extents are grid-searched in space $[0.001, 0.1]$ with step size $0.001$ until $10000$ random samples from the outer mesh surface have positive signed distance to the implicit surface and the same number of samples from the inner mesh surface have negative signed distance. For GIOM, mesh construction time take into account the whole process of bound computation, voxel trimming, and voxel union. Accordingly, mesh construction time for AdaptiveShells includes time spent on SDF grid computation and marching cubes mesh extraction.

Table 5: Ablation of fox object GIOM on resolution

| Res. | Time | I/O | Verticecs | Faces | Mem. | Max SD | Min SD | Mean D |
|---|---|---|---|---|---|---|---|---|
| $128^3$ | 38.7 | I | 1.12E+05 | 2.24E+05 | 3.8 | -2.12E-04 | -3.99E-02 | 5.77E-03 |
| | | O | 1.37E+05 | 2.75E+05 | 4.7 | 3.55E-02 | 5.02E-05 | 4.05E-03 |
| $256^3$ | 89.3 | I | 3.54E+05 | 7.09E+05 | 12.2 | -2.26E-05 | -1.42E-02 | 1.54E-03 |
| | | O | 3.95E+05 | 7.91E+05 | 13.6 | 1.12E-02 | 2.12E-05 | 1.18E-03 |
| $512^3$ | 159.5 | I | 6.47E+05 | 1.30E+06 | 22.2 | -1.38E-05 | -4.28E-03 | 1.02E-03 |
| | | O | 7.09E+05 | 1.42E+06 | 24.3 | 3.57E-03 | 9.61E-06 | 7.63E-04 |

Table 6: Ablation of fox object AdaptiveShells on resolution

| DE | Res. | Time | I/O | Vertices | Faces | Mem. | Max SD | Min SD | Mean D |
|---|---|---|---|---|---|---|---|---|---|
| 0.009 | $128^3$ | 0.3 | I | 1.20E+04 | 2.40E+04 | 0.4 | -3.29E-03 | -1.65E-02 | 9.33E-03 |
| | | | O | 1.63E+04 | 3.25E+04 | 0.6 | 1.41E-02 | 9.26E-04 | 8.72E-03 |
| 0.004 | $256^3$ | 1.9 | I | 5.38E+04 | 1.08E+05 | 1.8 | -1.14E-03 | -7.27E-03 | 4.08E-03 |
| | | | O | 6.14E+04 | 1.23E+05 | 2.1 | 6.59E-03 | 6.44E-04 | 3.92E-03 |
| 0.002 | $512^3$ | 14.1 | I | 2.25E+05 | 4.49E+05 | 7.7 | -7.02E-04 | -3.45E-03 | 2.02E-03 |
| | | | O | 2.40E+05 | 4.79E+05 | 8.2 | 3.21E-03 | 2.68E-04 | 1.98E-03 |
| 0.001 | $1024^3$ | 131.2 | I | 9.16E+05 | 1.83E+06 | 31.4 | -2.01E-04 | -1.69E-03 | 1.01E-03 |
| | | | O | 9.46E+05 | 1.89E+06 | 32.5 | 1.49E-03 | 3.22E-04 | 9.95E-04 |

Table 7: Ablation of tree object GIOM on resolution

| Res. | Time | I/O | Verticecs | Faces | Mem. | Max SD | Min SD | Mean D |
|---|---|---|---|---|---|---|---|---|
| $128^3$ | 54.4 | I | 1.39E+05 | 2.79E+05 | 4.8 | -4.16E-04 | -8.07E-02 | 1.12E-02 |
| | | O | 1.97E+05 | 3.94E+05 | 6.8 | 6.05E-02 | 7.69E-05 | 6.93E-03 |
| $256^3$ | 138.7 | I | 5.83E+05 | 1.17E+06 | 20.0 | -3.63E-05 | -4.64E-02 | 2.66E-03 |
| | | O | 6.34E+05 | 1.27E+06 | 21.8 | 4.96E-02 | 1.72E-05 | 1.76E-03 |
| $512^3$ | 299.1 | I | 1.31E+06 | 2.62E+06 | 45.0 | -6.68E-06 | -4.35E-02 | 1.24E-03 |
| | | O | 1.38E+06 | 2.77E+06 | 47.5 | 3.24E-02 | 4.72E-06 | 8.07E-04 |

Table 8: Ablation of tree object AdaptiveShells on resolution

| DE | Res. | Time | I/O | Vertices | Faces | Mem. | Max SD | Min SD | Mean D |
|---|---|---|---|---|---|---|---|---|---|
| 0.01 | $128^3$ | 0.4 | I | 1.52E+04 | 3.04E+04 | 0.5 | -3.07E-03 | -1.95E-02 | 1.05E-02 |
| | | | O | 2.40E+04 | 4.80E+04 | 0.8 | 1.87E-02 | 1.53E-03 | 9.59E-03 |
| 0.007 | $256^3$ | 2.7 | I | 6.80E+04 | 1.36E+05 | 2.3 | -1.25E-03 | -1.25E-02 | 7.11E-03 |
| | | | O | 9.24E+04 | 1.85E+05 | 3.2 | 1.36E-02 | 1.85E-04 | 6.89E-03 |
| 0.005 | $512^3$ | 19.8 | I | 2.88E+05 | 5.77E+05 | 9.9 | -9.57E-04 | -8.84E-03 | 5.03E-03 |
| | | | O | 3.58E+05 | 7.17E+05 | 12.3 | 8.05E-03 | 7.91E-04 | 4.97E-03 |
| 0.003 | $1024^3$ | 198.7 | I | 1.22E+06 | 2.44E+06 | 41.9 | -7.07E-04 | -5.25E-03 | 3.01E-03 |
| | | | O | 1.38E+06 | 2.77E+06 | 47.5 | 5.62E-03 | 6.93E-04 | 2.99E-03 |

Table 9: Ablation of koala object GIOM on resolution

| Res. | Time | I/O | Verticecs | Faces | Mem. | Max SD | Min SD | Mean D |
|---|---|---|---|---|---|---|---|---|
| $128^3$ | 47.5 | I | 1.83E+05 | 3.67E+05 | 6.3 | -8.20E-05 | -5.61E-02 | 2.73E-03 |
| | | O | 2.06E+05 | 4.12E+05 | 7.1 | 3.82E-02 | 3.55E-05 | 2.01E-03 |
| $128^3$ | 85.3 | I | 3.66E+05 | 7.33E+05 | 12.6 | -3.11E-05 | -1.73E-02 | 1.42E-03 |
| | | O | 4.07E+05 | 8.15E+05 | 14.0 | 1.28E-02 | 2.07E-05 | 1.07E-03 |
| $512^3$ | 124.4 | I | 5.25E+05 | 1.05E+06 | 18.0 | -2.56E-05 | -1.47E-01 | 1.34E-03 |
| | | O | 5.76E+05 | 1.15E+06 | 19.8 | 4.68E-03 | 1.18E-05 | 9.62E-04 |
| $1024^3$ | 165.3 | I | 6.37E+05 | 1.28E+06 | 21.9 | -6.42E-06 | -4.30E-03 | 1.27E-03 |
| | | O | 6.96E+05 | 1.39E+06 | 23.9 | 4.66E-03 | 1.02E-05 | 9.55E-04 |

Table 10: Ablation of koala object AdaptiveShells on resolution

| DE | Res. | Time | I/O | Vertices | Faces | Mem. | Max SD | Min SD | Mean D |
|---|---|---|---|---|---|---|---|---|---|
| 0.008 | $128^3$ | 0.3 | I | 2.68E+04 | 5.36E+04 | 0.9 | -2.51E-03 | -1.48E-02 | 8.15E-03 |
| | | | O | 2.98E+04 | 5.96E+04 | 1.0 | 1.36E-02 | 2.23E-04 | 7.85E-03 |
| 0.004 | $256^3$ | 2.1 | I | 1.11E+05 | 2.21E+05 | 3.8 | -1.45E-03 | -8.61E-03 | 4.04E-03 |
| | | | O | 1.17E+05 | 2.33E+05 | 4.0 | 6.94E-03 | 5.19E-04 | 3.96E-03 |
| 0.002 | $512^3$ | 14.9 | I | 4.49E+05 | 8.98E+05 | 15.4 | -3.33E-04 | -3.88E-03 | 2.01E-03 |
| | | | O | 4.61E+05 | 9.22E+05 | 15.8 | 3.21E-03 | 7.83E-05 | 1.99E-03 |
| 0.001 | $1024^3$ | 175.8 | I | 1.81E+06 | 3.62E+06 | 62.1 | -3.11E-04 | -1.81E-03 | 1.00E-03 |
| | | | O | 1.83E+06 | 3.67E+06 | 62.9 | 1.67E-03 | 1.45E-04 | 9.98E-04 |

Table 11: Ablation of cat object GIOM on resolution

| Res. | Time | I/O | Verticecs | Faces | Mem. | Max SD | Min SD | Mean D |
|---|---|---|---|---|---|---|---|---|
| $128^3$ | 36.6 | I | 1.22E+05 | 2.45E+05 | 4.2 | -6.07E-05 | -8.92E-02 | 3.60E-03 |
| | | O | 1.39E+05 | 2.78E+05 | 4.8 | 1.97E-02 | 3.12E-05 | 2.37E-03 |
| $256^3$ | 67.0 | I | 2.79E+05 | 5.58E+05 | 9.6 | -3.77E-05 | -8.56E-03 | 1.38E-03 |
| | | O | 3.11E+05 | 6.22E+05 | 10.7 | 7.89E-03 | 7.01E-06 | 1.01E-03 |
| $512^3$ | 100.1 | I | 4.14E+05 | 8.28E+05 | 14.2 | -1.24E-05 | -2.63E-02 | 1.18E-03 |
| | | O | 4.56E+05 | 9.13E+05 | 15.7 | 4.63E-03 | 8.62E-06 | 8.55E-04 |
| $1024^3$ | 123.3 | I | 5.07E+05 | 1.01E+06 | 17.4 | -2.28E-06 | -1.27E-02 | 1.15E-03 |
| | | O | 5.57E+05 | 1.11E+06 | 19.1 | 4.89E-03 | 6.67E-06 | 8.46E-04 |

Table 12: Ablation of cat object AdaptiveShells on resolution

| DE | Res. | Time | I/O | Vertices | Faces | Mem. | Max SD | Min SD | Mean D |
|---|---|---|---|---|---|---|---|---|---|
| 0.007 | $128^3$ | 0.3 | I | 1.41E+04 | 2.83E+04 | 0.5 | -4.16E-04 | -1.35E-02 | 7.23E-03 |
| | | | O | 1.64E+04 | 3.28E+04 | 0.6 | 1.26E-02 | 1.98E-03 | 6.79E-03 |
| 0.003 | $256^3$ | 1.9 | I | 5.94E+04 | 1.19E+05 | 2.0 | -4.17E-04 | -5.58E-03 | 3.05E-03 |
| | | | O | 6.33E+04 | 1.27E+05 | 2.2 | 5.44E-03 | 6.13E-04 | 2.94E-03 |
| 0.002 | $512^3$ | 14.0 | I | 2.41E+05 | 4.82E+05 | 8.3 | -7.44E-04 | -3.38E-03 | 2.01E-03 |
| | | | O | 2.51E+05 | 5.03E+05 | 8.6 | 3.41E-03 | 1.01E-03 | 1.99E-03 |
| 0.001 | $1024^3$ | 126.9 | I | 9.76E+05 | 1.95E+06 | 33.5 | -3.31E-04 | -1.60E-03 | 1.00E-03 |
| | | | O | 9.97E+05 | 1.99E+06 | 34.2 | 1.75E-03 | 4.46E-04 | 9.96E-04 |

Table 13: Ablation of skull object GIOM on resolution

| Type | Res. | Time | Vertices | Faces | Mem. | Max SD | Min SD | Mean D |
|------|------|------|----------|-------|------|--------|--------|--------|
| I | $128^3$ | 53.5 | 1.58E+05 | 3.15E+05 | 5.4 | -4.42E-04 | -1.06E-01 | 9.88E-03 |
| O |  |  | 2.26E+05 | 4.52E+05 | 7.8 | 1.08E-01 | 2.74E-04 | 9.49E-03 |
| I | $256^3$ | 140.3 | 6.82E+05 | 1.36E+06 | 23.4 | -5.32E-05 | -2.64E-02 | 2.29E-03 |
| O |  |  | 8.02E+05 | 1.60E+06 | 27.5 | 5.16E-02 | 2.85E-05 | 2.05E-03 |
| I | 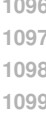$512^3$ | 232.2 | 1.74E+06 | 3.48E+06 | 59.7 | -1.16E-05 | -4.96E-03 | 9.31E-04 |
| O |  |  | 1.48E+06 | 2.95E+06 | 50.6 | 5.79E-03 | 1.77E-05 | 1.02E-03 |

Table 14: Ablation of skull object AdaptiveShells on resolution

| DE | Res. | Time | I/O | Vertices | Faces | Mem. | Max SD | Min SD | Mean D |
|----|------|------|-----|----------|-------|------|--------|--------|--------|
| 0.008 | $128^3$ | 0.4 | I | 1.97E+04 | 3.94E+04 | 0.7 | -3.40E-03 | -1.50E-02 | 8.21E-03 |
|  |  |  | O | 2.83E+04 | 5.65E+04 | 1.0 | 1.54E-02 | 3.44E-04 | 7.43E-03 |
| 0.004 | $256^3$ | 2.9 | I | 9.77E+04 | 1.95E+05 | 3.4 | -1.10E-03 | -6.84E-03 | 4.09E-03 |
|  |  |  | O | 1.09E+05 | 2.19E+05 | 3.8 | 6.91E-03 | 1.87E-04 | 3.77E-03 |
| 0.002 | $512^3$ | 21.9 | I | 4.10E+05 | 8.20E+05 | 14.1 | -7.19E-04 | -3.20E-03 | 2.03E-03 |
|  |  |  | O | 4.31E+05 | 8.63E+05 | 14.8 | 3.33E-03 | 5.55E-04 | 1.93E-03 |
| 0.001 | $1024^3$ | 191.7 | I | 1.67E+06 | 3.34E+06 | 57.2 | -4.82E-04 | -1.58E-03 | 1.01E-03 |
|  |  |  | O | 1.71E+06 | 3.42E+06 | 58.7 | 1.76E-03 | 3.89E-04 | 9.81E-04 |

Table 15: Ablation of lion statue object GIOM on resolution

| Type | Res. | Time | Vertices | Faces | Mem. | Max SD | Min SD | Mean D |
|------|------|------|----------|-------|------|--------|--------|--------|
| I | $128^3$ | 96.8 | 4.16E+05 | 8.32E+05 | 14.3 | -4.68E-04 | -1.94E-01 | 6.49E-03 |
| O |  |  | 4.75E+05 | 9.49E+05 | 16.3 | 5.13E-02 | 4.15E-04 | 4.42E-03 |
| I | $256^3$ | 282.8 | 1.42E+06 | 2.85E+06 | 48.9 | -1.35E-04 | -1.97E-02 | 1.30E-03 |
| O |  |  | 1.52E+06 | 3.04E+06 | 52.2 | 1.38E-02 | 1.34E-04 | 1.16E-03 |
| I | $512^3$ | 525.3 | 2.45E+06 | 4.91E+06 | 84.2 | -2.29E-05 | -5.71E-03 | 1.09E-03 |
| O |  |  | 2.56E+06 | 5.13E+06 | 88.0 | 4.25E-03 | 1.72E-05 | 9.16E-04 |

Table 16: Ablation of lion statue object AdaptiveShells on resolution

| DE | Res. | Time | I/O | Vertices | Faces | Mem. | Max SD | Min SD | Mean D |
|----|------|------|-----|----------|-------|------|--------|--------|--------|
| 0.008 | $128^3$ | 0.5 | I | 5.90E+04 | 1.18E+05 | 2.0 | -1.76E-03 | -1.61E-02 | 8.11E-03 |
|  |  |  | O | 6.24E+04 | 1.25E+05 | 2.1 | 1.39E-02 | 6.92E-04 | 7.93E-03 |
| 0.003 | $256^3$ | 2.7 | I | 2.42E+05 | 4.85E+05 | 8.3 | -3.10E-04 | -5.80E-03 | 3.02E-03 |
|  |  |  | O | 2.47E+05 | 4.95E+05 | 8.5 | 5.68E-03 | 5.34E-04 | 2.98E-03 |
| 0.002 | $512^3$ | 17.8 | I | 9.75E+05 | 1.95E+06 | 33.5 | -8.70E-04 | -3.32E-03 | 2.00E-03 |
|  |  |  | O | 9.88E+05 | 1.98E+06 | 33.9 | 3.06E-03 | 6.82E-04 | 1.99E-03 |
| 0.001 | 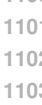$1024^3$ | 151.5 | I | 3.92E+06 | 7.83E+06 | 134.5 | -5.48E-04 | -1.44E-03 | 1.00E-03 |
|  |  |  | O | 3.94E+06 | 7.88E+06 | 135.3 | 1.50E-03 | 5.36E-04 | 9.99E-04 |

Table 17: Ablation of scorpion object GIOM on resolution

| Type | Res. | Time | Vertices | Faces | Mem. | Max SD | Min SD | Mean D |
|------|------|------|----------|-------|------|--------|--------|--------|
| $128^3$ | 55.3 | I | 2.33E+05 | 4.67E+05 | 8.0 | -8.27E-05 | -1.07E-01 | 7.38E-03 |
| | | O | 2.91E+05 | 5.83E+05 | 10.0 | 4.22E-02 | 7.87E-05 | 4.90E-03 |
| $256^3$ | 150.7 | I | 8.06E+05 | 1.61E+06 | 27.7 | -5.10E-05 | -1.46E-01 | 2.13E-03 |
| | | O | 8.95E+05 | 1.79E+06 | 30.7 | 1.07E-02 | 2.40E-05 | 1.31E-03 |
| $512^3$ | 320.6 | I | 1.56E+06 | 3.12E+06 | 53.6 | -1.04E-05 | -3.95E-02 | 1.09E-03 |
| | | O | 1.70E+06 | 3.40E+06 | 58.4 | 3.89E-03 | 1.29E-05 | 7.61E-04 |

Table 18: Ablation of scorpion object AdaptiveShells on resolution

| DE | Res. | Time | I/O | Vertices | Faces | Mem. | Max SD | Min SD | Mean D |
|----|------|------|-----|----------|-------|------|--------|--------|--------|
| 0.007 | $128^3$ | 0.4 | I | 2.79E+04 | 5.57E+04 | 1.0 | -1.27E-03 | -1.41E-02 | 7.42E-03 |
| | | | O | 3.51E+04 | 7.02E+04 | 1.2 | 1.49E-02 | 3.15E-04 | 6.66E-03 |
| 0.004 | $256^3$ | 3.0 | I | 1.19E+05 | 2.38E+05 | 4.1 | -1.23E-03 | -7.23E-03 | 4.10E-03 |
| | | | O | 1.35E+05 | 2.71E+05 | 4.6 | 6.65E-03 | 6.51E-04 | 3.91E-03 |
| 0.002 | $512^3$ | 22.1 | I | 4.94E+05 | 9.88E+05 | 17.0 | -7.28E-04 | -3.77E-03 | 2.02E-03 |
| | | | O | 5.27E+05 | 1.05E+06 | 18.1 | 3.45E-03 | 5.90E-04 | 1.98E-03 |
| 0.001 | $1024^3$ | 186.2 | I | 2.02E+06 | 4.03E+06 | 69.2 | -3.18E-04 | -1.61E-03 | 1.01E-03 |
| | | | O | 2.08E+06 | 4.16E+06 | 71.5 | 1.51E-03 | 3.09E-04 | 9.94E-04 |

## C.2 EXPERIMENT DETAILS

### C.2.1 REAL-TIME RENDERING

**Task Setup and Metrics.**   Given a pre-constructed Neural SDF and specified camera settings (position, direction, and front-of-view (FOV) angle), our goal is to render high-resolution, high-fidelity images corresponding to the input geometry. We evaluate our method using four pretrained neural SDFs representing the fox, cat, koala, and tree models. Each SDF consists of 8 layers of width 64 (tree) or 32 (others). The fox SDF is directly inherited from the open-source implementation in Sharp and Jacobson (2022), while the remaining models were trained on meshes from Stein (2024). For rendering, we use the neural SDF as the exact geometric representation, and employ the bounding meshes solely to accelerate ray casting. Specifically, we (1) find the intersection between each ray and the implicit surface with Alg. 2, and (2) compute the surface normal by querying the neural SDF at the intersection point. We used PyTorch Paszke et al. (2019) for MLP inference and Optix NVIDIA Corporation (2024) for the ray-mesh intersection. To quantitatively evaluate rendering quality, we render 50 images per object at a resolution of $1024 \times 1024$ from 50 fixed camera positions spanning the surface of a sphere centered on the object. We report SSIM, PSNR, and RMSE as image quality metrics, along with FPS to evaluate rendering efficiency.

**Baselines and Experiment Details.**   We compare against four baselines: sphere tracing Hart (1996), affine-arithmetic-based interval tracing Sharp and Jacobson (2022), Adaptive Shells for SDFs, and 0-level set approximation via marching cubes. Sphere tracing marches rays by the signed distance evaluated at the ray head in each iteration and terminates when the absolute value falls below a small threshold. It can cause the ray to graze near a complex surface, wasting compute on excessive samples. Interval tracing adjusts step size dynamically based on whether the next step is safe, which is determined by the bounds computed via affine arithmetic. Although this approach guarantees precision when the neural SDF is not exact, it still takes more steps than necessary, especially into the empty space. Nonetheless, it is a state-of-the-art approach for rendering with *guaranteed* precision. Adaptive shells are originally used for accelerating queries on NeRF, but we use them here to denote inner and outer mesh extraction via dilation and erosion with marching cubes. The dilation and erosion extents are controlled by the varying kernel size of Neus in the original work, accommodating both solid and fluffy surfaces. Since we only use simple MLPs to encode uniformly solid surfaces, we manually tune the dilation and erosion extents until the bounding meshes are sound and combine them with Alg. 2 as our third baseline. The last baseline, 0 level set approximation via marching cubes (0 level MC), is a classic yet error-prone approach. We can approximate the ray-surface intersection with ray-mesh intersection, but the inherent lack in flexibility of marching cubes can often introduce missing volumes or artifacts, thereby obscuring the rendering result.

We adaptively split for at least 27 rounds for each object to construct GIOM and GIOM-Z, using an early stop distance threshold of 0.001. We apply the same threshold to any baseline that it may concern, such as sphere and interval tracing. The same value is used for the precision $\delta$ of Alg. 2.

### C.2.2 Physics Simulation

**Task Setup and Metrics.** We run particle collision and cloth simulation to demonstrate two dimensions of the benefits of GIOM in physics simulation: efficiency and quality. In particle collision tasks, our goal is to efficiently detect collision between the query body and tens of thousands of small spherical particles. We drop a $64 \times 64 \times 16$ particle grid on each object under gravity $g = 9.81$ from 2.5 meters above the ground, then we measure the time spent on collision detection only. In cloth simulation tasks, we drop a piece cloth modeled as a $30 \times 50$ 2-manifold particle grid onto a object and inspect the authenticity of the simulation. We run both simulation tasks with NVIDIA Warp Macklin (2022). For the mesh-SDF continuous collision detection, we use Pelletier-Guénette et al. (2025) as the baseline and enhance its broad phase triangle culling with GIOM. Given a dynamic mesh colliding with a static SDF, our goal is to improve collision detection efficiency without loss of robustness. We measure the time spent o collision detection as well as the number of triangles tested in the narrow phase. All visual results are rendered with Blender Community (2023).

**Baselines and Experiment Details.** For particle collision, we compare our GIOM against two baselines: voxel SDF and Axis-Aligned Bounding Box (AABB). Voxel SDF can be directly sampled from neural SDF. Querying the former with the point involves only trilinear interpolation and saves the expensive call on a neural network, boosting efficiency significantly. AABB can be used to further decrease bandwidth by filtering points that can potentially collide with the query body. For cloth simulation, we compare against two baselines: voxel SDF with Euler integrator, a real-time but less accurate approach, and voxel SDF with XPBD integrator, a slow but refined approach. We adaptively split for up to 15 iterations to keep the bounding mesh simple because complex bounding mesh can increase the filtering overhead. For both tasks, we simulate for 60 frames per second and 64 substeps per frame. The cloth grid has cell dimension $0.05 \times 0.05$ and kinematics coefficients $k_e = k_a = 2.5e2, k_d = 1.0e1$. For mesh-SDF continuous collision detection, we used 60 simulation steps per second with a friction coefficient of 0.45.

### C.2.3 Constructive Solid Geometry

**Tasks Setup and Metrics.** Given two SDFs, our goal is to reconstruct the signed distance field of their CSG. The signed distance field should not only model the new border correctly but also be exact everywhere in the space. We measure reconstruction time (in minutes) and quality (with L1-error) as well as the time to query the reconstructed SDF (in milliseconds).

**Baselines and Experiment Details.** We compare our method against CSG-nSDF Marschner et al. (2023a). CSG-nSDF is a state-of-the-art approach that models the CSG of SDFs with a neural SDF. By integrating critical geometric objectives into the training loss function, it remains one of the most accurate approximation of the exact signed distance field of CSG. We use the open-source CSG model on the union of a square and a circle. We adaptively split for 18 iterations for the square and the circle. We accelerate closest point queries with Warp kernels.

## C.3  ADDITIONAL FIGURES

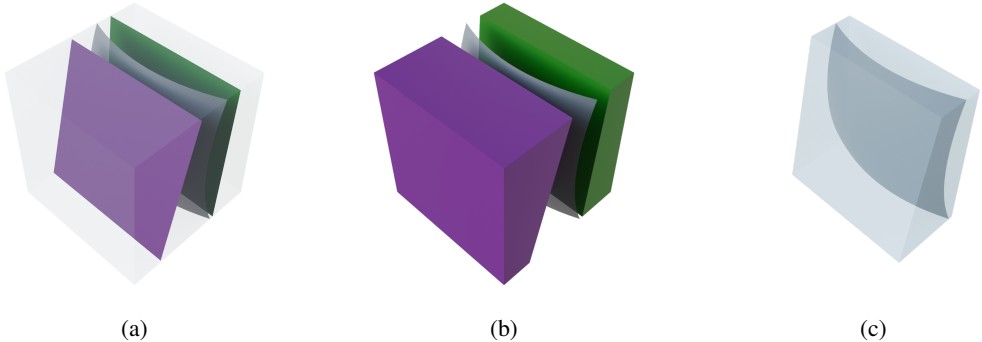

(a)                                    (b)                                    (c)

Figure 9: Voxel trimming with inner (green) and outer (purple) bounding planes of a local surface (gray) in Figure 9a. The inner and outer volumes are visualized as purple and green convex polytopes 9b. The region (light blue) between the bounding planes conservatively contains the local isosurface in Figure 9c (i.e. $\mathcal{V}^{\mathcal{C}}_+ \setminus \mathcal{V}^{\mathcal{C}}_- \supset \mathcal{S}^{\mathcal{C}}$)

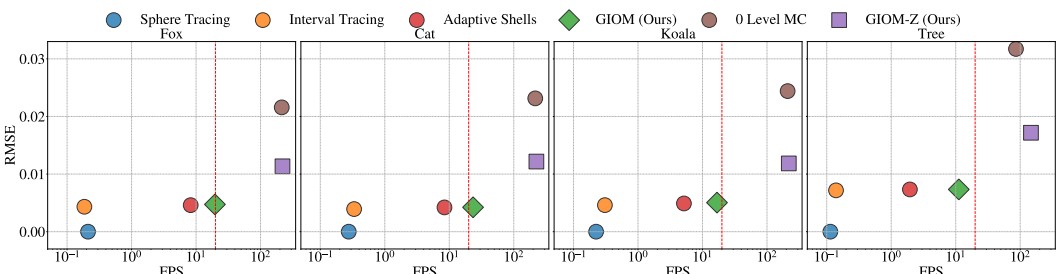

Figure 10: **Rendering quality (RMSE) vs speed (FPS) trade-off** among all competing algorithms.

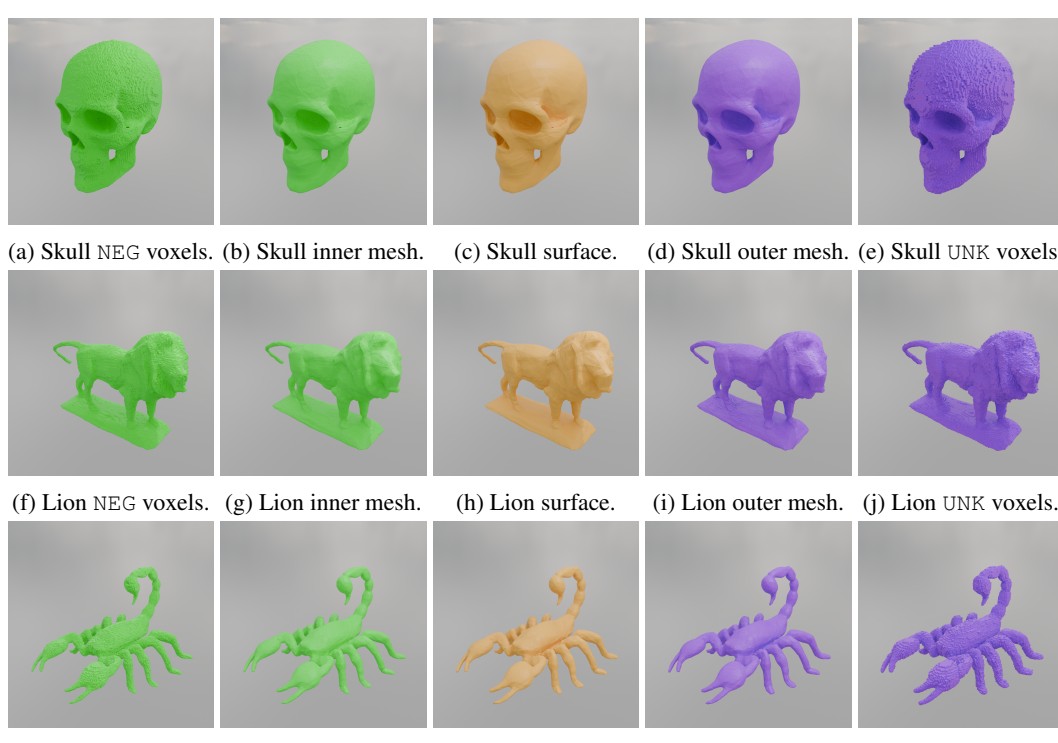

(a) Skull `NEG` voxels. (b) Skull inner mesh. (c) Skull surface. (d) Skull outer mesh. (e) Skull `UNK` voxels.

(f) Lion `NEG` voxels. (g) Lion inner mesh. (h) Lion surface. (i) Lion outer mesh. (j) Lion `UNK` voxels.

(k) Scor. `NEG` voxels. (l) Scor. inner mesh. (m) Scor. surface. (n) Scor. outer mesh. (o) Scor. `UNK` voxels.

Figure 11: **Visualization of GIOM with max resolution $512^3$.** GIOM can produce sound inner and outer bounding meshes at medium resolutions. The bounding meshes not only align well with the geometric details of the bounded surfaces but are also tight—the geometric difference between the inner and outer meshes is barely noticeable by human eyes. Thanks to voxel trimming 3.2, the inner and outer meshes are visibly refined from `NEG` and `UNK` voxels. For geometries involving uneven surfaces (lion statue), holes (skull), and narrow structures (scorpion), GIOM can still delivery consistent performance. The surfaces (yellow) are approximated marching cubes with resolution $1024^3$.

