# OpenReview forum: "Guaranteed Bounding Meshes Extraction from Neural Implicit Surfaces via Neural Network Verification"
_ICLR.cc/2026/Conference — ICLR 2026 Conference Withdrawn Submission_

### Official Review · Reviewer_pNRh · 2025-10-26

**Soundness:** 3
**Presentation:** 4
**Contribution:** 3
**Rating:** 8
**Confidence:** 3

**Summary:**

This paper proposes Guaranteed Inner-and-Outer Meshes (GIOM), a new algorithm for computing bounding meshes for neural implicit surfaces (especially neural signed distance functions (SDFs)). They treat SDF queries as neural network verification problems, enabling the use of CROWN, a linear bound propagation method, to compute affine upper and lower bounds on the SDF output over voxel regions. As a result, the method can obtain an explicit mesh representation that (1) guarantees correctness, (2) adapts to surface complexity, and (3) integrates directly with traditional mesh-based rendering and physics engines.

GIOM is demonstrated on three tasks: neural rendering, collision detection, and constructive solid geometry (CSG). It shows up to 3× faster rendering, 5× faster physics simulation, and 10× lower reconstruction error in CSG compared to baselines such as Adaptive Shells, Spelunking the Deep, and CSG-nSDF.

**Strengths:**

- The paper introduces a novel perspective by framing geometric queries on neural implicit surfaces as neural network verification problems, which in turn leads to the subsequent proposed GIOM algorithm.
- The paper is technically sound. It offers a clear mathematical derivation of how affine upper and lower bounds (from CROWN-style linear bound propagation) can be used to construct certified inner and outer meshes.
- The empirical results are strong. It demonstrates three applications in neural rendering, physics-based collision detection, and constructive solid geometry (CSG), and achieves significant performance gains in all cases.
- Finally, the paper is also well-written and easy to follow.

**Weaknesses:**

- While the paper reports timing metrics such as FPS for rendering and milliseconds per frame for physics simulation, it omits the time required to construct the inner and outer meshes. When the underlying network of the implicit surfaces is MLPs, which is very common, each voxel’s bound computation entails a full backpropagation through the entire network. This step can become computationally expensive if the network is deep or if many voxels require refinement, and may result in higher costs in total than direct forward queries.

- The idea of representing an object as multiple surfaces is not rare in literature and can be further discussed in the related works. For example, in [1][2][3].

- If I am understanding correctly, CROWN’s linear relaxation could still potentially result in loose bounds (maybe for highly nonlinear or deep architectures), and there are only theoretical guarantees for the correctness of the bounds but not the tightness. It would be valuable to include an ablation showing how the error varies with subdivision depth.

- In addition, no explicit failure cases are shown where GIOM’s bounds might degrade or where the linear approximation becomes visually noticeable. A discussion of such scenarios would improve completeness without undermining the paper’s overall contributions.

[1] Esposito et. al. Volumetric Surfaces: Representing Fuzzy Geometries with Layered Meshes

[2] Wang et. al. A Simple Approach to Differentiable Rendering of SDFs

[3] Seyb etl. al. From microfacets to participating media: A unified theory of light transport with stochastic geometry

**Questions:**

- Do the MLPs used in the paper include positional encodings (e.g. fourier or sinusoidal embedding)? If not, supplementing these features would be important for evaluating real-case implicit surfaces.

- Could the authors report the actual time and memory consumption for the construction of the meshes before downstream applications?

- How does GIOM compare in terms of time and memory with classical bounding-volume hierarchies such as octrees or BVHs, which also subdivide space adaptively?

- Can the authors supplement and discuss some failure cases?

---

> ### Author Response · Authors · 2025-11-29
> **Thank you for the valuable feedback and we have addressed your questions and concerns below**
>
> Dear Reviewer pNRh,
>
> Thank you for acknowledging the importance of our work and offering constructive feedback! We address your questions and concerns point by point below:
>
> > **W1, W3, Q2, Q3** What is the bounding mesh construction cost and how does it compare with other bounding volume hierarchies?
>
> **Re.:** We have updated the paper with Table 5—18 in Appendix that present the time and memory consumptions of GIOM compared with Adaptive Shells (dilation/erosion plus marching cubes).  We use Adaptive Shells as a strong substitute for classical octree and BVHs because Adaptive Shells can yield bounding meshes that are comparably tight as GIOM, which add refinement based on a KD-tree structure. Below we report the average performance across seven objects and compare with Adaptive Shells:
>
> **GIOM (Ours)**
>
> |   Resolution | Time          | Memory      | Mean Distance       |
> |--------------|---------------:|-------------:|---------------------:|
> |      $128^3$ | 53.0 ± 20.5   | 14.8 ± 7.7  | 5.95E-03 ± 3.01E-03 |
> |      $256^3$ | 131.3 ± 73.3  | 46.0 ± 28.0 | 1.60E-03 ± 4.59E-04 |
> |      $512^3$ | 251.6 ± 147.2 | 85.6 ± 51.2 | 9.97E-04 ± 8.30E-05 |
>
> **Adaptive Shells**
>
> |   Resolution | Time         | Memory       | Mean Distance       |
> |--------------|--------------:|--------------:|---------------------:|
> |      $128^3$ | 0.4 ± 0.1    | 1.9 ± 1.0    | 8.25E-03 ± 1.05E-03 |
> |      $256^3$ | 2.5 ± 0.5    | 7.9 ± 4.1    | 4.37E-03 ± 1.41E-03 |
> |      $512^3$ | 18.3 ± 3.6   | 31.6 ± 16.3  | 2.50E-03 ± 1.07E-03 |
> |     $1024^3$ | 166.6 ± 27.3 | 126.9 ± 65.3 | 1.38E-03 ± 7.45E-04 |
>
> To ensure **empirical soundness** for Adaptive Shells, we incrementally adjust the dilation/erosion thicknesses until:
> 1. randomly-sampled points around the outer mesh have strictly positive signed distances, and
> 2. those around the inner mesh have strictly negative signed distances.
>
> This ensures no observed violations but provides **no formal guarantees**.
> GIOM, in contrast, enforces **theoretical soundness** for the entire mesh surface. Under the same resolution (e.g., $512^3$), GIOM **improves tightness**, measured by the mean distance from random samples on the bounding meshes to the implicit surface. For similar tightness (e.g., GIOM $512^3$ vs. Adaptive Shells $1024^3$), GIOM **reduces memory cost** despite higher preprocessing time.
>
> > **W2** The idea of representing an object as multiple surfaces is not rare in literature and can be further discussed in the related works [1][2][3].
>
> **Re.:** Although the suggested works also explore representations of an object as multiple surfaces, they focus on improving different facets of rendering, such as volumetric representation and differentiability, and are orthogonal to our method. However, we seriously consider your suggestion to discuss more works with high relevance in the related works section and have included all the aforementioned papers in the reference.
>
> > **W4, Q1, Q4** Are there any failure cases, and do the MLPs used in the paper include positional encodings?
>
> **Re.:** The linear approximation can be noticeable with networks that include positional encoding. It takes more time to split the voxels to achieve tight bounding meshes on those networks. We present the results in the appendix (Tables 13—16 and the first two rows of Figure 11) by bounding two MLPs with positional encoding that models the lion statue and skull. While the bounding meshes do not become extremely loose, the processing cost can become large (e.g., more than 8 minutes to construct GIOM on the skull with resolution $512^3$).
>
> [1] Esposito et. al. Volumetric Surfaces: Representing Fuzzy Geometries with Layered Meshes
>
> [2] Wang et. al. A Simple Approach to Differentiable Rendering of SDFs
>
> [3] Seyb etl. al. From microfacets to participating media: A unified theory of light transport with stochastic geometry

---

### Official Review · Reviewer_JhSZ · 2025-10-27

**Soundness:** 1
**Presentation:** 2
**Contribution:** 1
**Rating:** 2
**Confidence:** 5

**Summary:**

The paper proposes to represent geometric queries for neural signed distance functions as verification problems using linear bound propagation. The objective is to compute inner and outer bounding meshes that contain the zero-level set. It does that by performing a probably very costly grid-based preprocessing (to be verified in the rebuttal discussion). The process consists of:

1) Voxelizaton;
2) Computation of two polyhedral volumes (positive and negative) for each voxel;
2.1) Classification of voxels as positive, negative or unknown based on linear bound propagation via CROWN;
2.2) Voxel trimming by reusing the linear bounds and performing a subdivision scheme;
2.3) Union of all positive, negative, and unknown voxels: the polyhedral meshes are merged.

The paper also proposes a way to extract the zero-level set mesh based on the bounding meshes and shows applications in rendering, collision detection, and CSG operation.

**Strengths:**

The presentation is its stronger trait. Even though an overview image of the method is missing, it is easy to follow.

**Weaknesses:**

## The contribution is incremental:

Spelunking the Deep [1] first established that geometric queries on neural implicit representations can be framed as range-bound computations over the activation domain, yielding certified guarantees via affine arithmetic.
The proposed method extends this same conceptual formulation but replaces affine-arithmetic bounds with linear bound propagation techniques from neural network verification (CROWN-style relaxations), leading to tighter yet analogous certified bounds. While this substitution improves tightness, it remains largely consistent with the prior framework. The contribution is therefore incremental in conceptual novelty, driven mainly by deploying a more advanced bound-propagation backend.

## Motivation for trading SDFs for boundary meshes

The approach proposed in the paper has an important drawback in comparison with directly applying the neural implicit SDF for the geometric queries. As pointed by the fact that the approach does not impose an eikonal regularization and that it computes the normals for rendering using finite differences, the representation loses higher order differentiability. Being able to analytically (or with autodiff) compute differential surface properties is one of the most appealing traits of neural SDFs. It is possible to compute normals and curvature directly from a SIREN [2] SDF. Grid-based approaches struggle to maintain that differentiability because they rely on voxel interpolation. If those properties are lost in the process, why would we care about training a neural SDF from a point cloud in the first place?

## The preprocessing is not evaluated

As discussed in the Summary, the preprocessing of the proposed method is extensive. I am skeptical that its cost is not prohibitive. The paper mentions efficiency but I could not find any evaluation in this regard. Even if this pipeline focuses on post-training certified performance, an end-to-end runtime comparison against fast neural implicit methods (e.g., SIREN, Instant-NGP, NGLOD, BACON) would help contextualize the practicality of the approach. Shallow SIRENs are a good initial baseline (SIRENs only work with a small number of hidden layers because of the frequency composition generated by the successive multiplications of the periodic activation functions, which may introduce noise as the number of hidden layers increase). Given that the proposed method is a grid-based approach, Instant-NGP [3] is also a candidate comparison for the rendering task. It is unclear whether the proposed approach could surpass those methods since SIRENs train very fast and Instant-NGP trains instantaneously.

## The evaluation lacks diversity

Developing a little bit more on the discussion started about the preprocessing, the paper followed an evaluation approach similar from Spelunking the Deep. However, such evaluation does not sufficiently demonstrate robustness or scalability to diverse real-world shapes.. The presented examples are very simple. For more challenging examples, please see NGLOD and Instant-NGP. They evaluate the Stanford Dataset and a subset of Thingy-10k, a more diverse set of examples.

The presented average numbers can hide problems on specific SDFs. Showing per-SDF metrics would strengthen the evaluation.

Additional metrics are missing. At least the chamfer distance of the generated meshes should be presented.

## Assessment

The current weaknesses are essential and numerous to be fixed in a rebuttal. This is why I recommend rejection.

## References

[1] Sharp, Nicholas, and Alec Jacobson. "Spelunking the deep: Guaranteed queries on general neural implicit surfaces via range analysis." ACM Transactions on Graphics (TOG) 41.4 (2022): 1-16.

[2] Sitzmann, Vincent, et al. "Implicit neural representations with periodic activation functions." Advances in neural information processing systems 33 (2020): 7462-7473.

[3] Müller, Thomas, et al. "Instant neural graphics primitives with a multiresolution hash encoding." ACM transactions on graphics (TOG) 41.4 (2022): 1-15.

[4] Takikawa, Towaki, et al. "Neural geometric level of detail: Real-time rendering with implicit 3d shapes." Proceedings of the IEEE/CVF conference on computer vision and pattern recognition. 2021.

[5] Lindell, David B., et al. "Bacon: Band-limited coordinate networks for multiscale scene representation." Proceedings of the IEEE/CVF conference on computer vision and pattern recognition. 2022.

**Questions:**

No additional questions.

---

> ### Author Response · Authors · 2025-11-29
> **Response (1/3)**
>
> Dear Reviewer JhSZ,
>
> We appreciate your valuable feedback and comments. We feel that most of the concerns are largely misunderstandings of our work, and we provide clarifications and supporting results below to address these points:
>
> > **The contribution is incremental.** GIOM remains largely consistent with the prior framework (Spelunking the Deep [1]). The contribution is therefore incremental in conceptual novelty, driven mainly by deploying a more advanced bound-propagation backend.
>
> **Re.:** We respectfully disagree that our work is highly incremental. Our work does not simply replace affine arithmetic with CROWN. GIOM not only benefits from the tighter numerical bounds but also leverages the linear bounds from CROWN. One of our main contributions is identifying the **connection between linear bounds and geometric primitives** and utilizing it to generate provably robust bounding meshes. Compared with the KD-Tree spatial hierarchy, the bounding meshes are visibly tighter (Appendix C.3 Figure 11) and enable new applications. For example, [1] must compute interval bounds on the fly to perform ray casting, but GIOM allows direct sampling inside the certified bounded region, significantly improving efficiency.
> We also **refined the splitting algorithm from [1] with the distance between bounding planes**. As shown in the table below, a naïve adaptive split (w/o guidance) leads to near-exponential growth in the number of nodes when the depth increases. In contrast, distance-guided splitting **scales almost linearly**, reducing node count by up to **4x** at deeper levels. This efficiency is crucial for enabling **high-resolution GIOM** without exponential time and memory cost.
> | Tree Depth | 1–17 | 18 | 19 | 20 | 21 | 22 | 23 | 24 | 25 | 26 | 27 |
> |-----------|:----:|---:|----:|-----:|------:|-------:|-------:|--------:|--------:|--------:|--------:|
> | **w/ Distance Guidance**  | same | 7132 | 9560 | 14750 | 21484 | 26130 | 34932 | 40420 | 40446 | 44790 | 32716 |
> | **w/o Distance Guidance** | same | 7132 | 9566 | 14828 | 22356 | 31498 | 49998 | 77284 | 114490 | 185984 | 146808 |
>
> Based on your constructive comments, we have updated the contribution list in Section 1 **Introduction** of our paper to emphasize our contribution in significantly improving the tightness and scalability of certified bounding volume with linear bounds.
>
> > **Motivation for trading SDFs for meshes.** GIOM does not impose Eikonal regularization and computes surface normal with finite difference.
>
> **Re.:** Thank you for the insightful comments, but we believe there might be some misunderstanding. Throughout the paper, we never claimed that we should trade SDFs for meshes. Instead, we use the bounding meshes together with SDFs to complement the drawbacks of the latter (i.e. being expensive to query and lacking connection to spatial hierarchies) while retaining its advantages in encoding normals and being differentiable. In our rendering algorithm, after finding the first-hit locations of the rays, we compute the surface normal by querying the neural SDFs rather than voxel interpolation. We primarily use finite-difference method for real-time efficiency, but the exact normal can always be computed by taking the gradient of the neural SDF at the hit-location.
>
> We have updated the relevant details in Appendix C.2.1 (**Task Setup and Metrics**) to avoid further confusion.

---

> ### Author Response · Authors · 2025-11-29
> **Response (2/3)**
>
> > **Preprocessing is not evaluated.** Preprocessing cost can be prohibitive. Cannot find evaluation on efficiency. Need to compare with fast neural implicit approaches like SIREN [2], Instant-NGP [3], NGLOD [4], and BACON [5].
>
> **Re.:** We recognize the lack of evaluation on the time and memory cost of preprocessing as a drawback of our paper and have updated Appendix C (Table 5—18) in our latest revision to fix it. We agree that efficiency is a favorable trait for any preprocessing method, but GIOM does not aim to outperform fast training pipelines such as Instant-NGP. Our focus is certified post-training robustness, complementary to these methods. With Table 5—18 in Appendix C showing that the preprocessing costs are not prohibitive, we argue that they are worth the robustness guarantee, which are critical in applications, such as real-time rendering, where the lack of such guarantees can lead to missing volumes (Figure 5 in our latest revision). Below we report the average preprocessing performance across seven objects and compare with Adaptive Shells:
>
> **GIOM (Ours)**
>
> |   Resolution | Time (s)          | Memory (MB)     | Mean Distance       |
> |--------------|---------------:|-------------:|---------------------:|
> |      $128^3$ | 53.0 ± 20.5   | 14.8 ± 7.7  | 5.95E-03 ± 3.01E-03 |
> |      $256^3$ | 131.3 ± 73.3  | 46.0 ± 28.0 | 1.60E-03 ± 4.59E-04 |
> |      $512^3$ | 251.6 ± 147.2 | 85.6 ± 51.2 | 9.97E-04 ± 8.30E-05 |
>
> **Adaptive Shells**
>
> |   Resolution | Time (s)        | Memory (MB)      | Mean Distance       |
> |--------------|--------------:|--------------:|---------------------:|
> |      $128^3$ | 0.4 ± 0.1    | 1.9 ± 1.0    | 8.25E-03 ± 1.05E-03 |
> |      $256^3$ | 2.5 ± 0.5    | 7.9 ± 4.1    | 4.37E-03 ± 1.41E-03 |
> |      $512^3$ | 18.3 ± 3.6   | 31.6 ± 16.3  | 2.50E-03 ± 1.07E-03 |
> |     $1024^3$ | 166.6 ± 27.3 | 126.9 ± 65.3 | 1.38E-03 ± 7.45E-04 |
>
> To ensure **empirical soundness** for Adaptive Shells, we incrementally adjust the dilation/erosion thicknesses until:
> 1. randomly-sampled points around the outer mesh have strictly positive signed distances, and
> 2. those around the inner mesh have strictly negative signed distances.
>
> This ensures no observed violations but provides **no formal guarantees**.
> GIOM, in contrast, enforces **theoretical soundness** for the entire mesh surface. Under the same resolution (e.g., $512^3$), GIOM **improves tightness**, measured by the mean distance from random samples on the bounding meshes to the implicit surface. For similar tightness (e.g., GIOM $512^3$ vs. Adaptive Shells $1024^3$), GIOM **reduces memory cost** despite higher preprocessing time.

---

> ### Author Response · Authors · 2025-11-29
> **Response (3/3)**
>
> > **Evaluation lacks diversity.** Evaluation does not sufficiently demonstrate robustness or scalability to diverse real-world shapes. The presented average numbers can hide problems on specific SDFs. Showing per-SDF metrics would strengthen the evaluation. Need additional metrics on the generated bounding meshes.
>
> **Re.:** We have updated Appendix C (Tables 13–18 and Figure 11) in our latest revision to include more complex shapes, such as the skull (with holes), lion statue (uneven surface features), and scorpion (thin, long structures). We fitted the new shapes with MLPs with 8 layers and 64 neurons in each hidden layer. For the skull and lion statue, we added positional encoding, a widely-adopted trick to fit complex, real-world surfaces, with 10 frequencies (from $2^-3$ to $2^6$) to enhance neural SDF accuracy. We list GIOM performance at resolution $512^3$ for all implicit surfaces included in the paper in the ascending order of visual complexity to demonstrate scalability.
>
> | Object      |   Time |   Memory |   Mean Distance |
> |-------------|--------:|----------:|-----------------:|
> | Koala       |  124.4 |     37.8 |       1.15E-03  |
> | Cat         |  100.1 |     29.9 |      1.01E-03 |
> | Fox         |  159.5 |     46.5 |       8.91E-04 |
> | Tree        |  299.1 |     90.2 |       1.01E-03  |
> | **Scorpion**    |  320.6 |    112.0   |       9.25E-04 |
> | **Lion Statue** |  232.2 |    110.3 |       9.75E-04 |
> | **Skull**       |  525.3 |    172.2 |       1.00E-03 |
>
> For complex objects, GIOM is still tight (with low mean distance) under low memory budgets compared to Adaptive Shells despite being more time consuming. We show the performance of Adaptive Shells at resolution $1024^3$ with comparable tightness below for reference.
>
> | Object      |   Time |   Memory |   Mean Distance |
> |-------------|--------:|----------:|-----------------:|
> | Koala       |  175.8 |    125.0   |      9.99E-04  |
> | Cat         |  126.9 |     67.7 |       9.98E-04  |
> | Fox         |  131.2 |     63.9 |       1.00E-03  |
> | Tree        |  198.7 |     89.4 |       3.00E-03     |
> | **Scorpion**    |  186.2 |    140.7 |       1.00E-03  |
> | **Lion Statue** |  191.7 |    115.9 |       9.95E-04 |
> | **Skull**       |  151.5 |    269.8 |       9.99E-04 |
>
> Regarding per-SDF metrics, we already have those in the original submission: for rendering in Appendix C (Figure 9) and for physics simulations in Table 3.
>
> We have included additional evaluations on the tightness of GIOM in Appendix C as mentioned above. We included the range of signed distance and mean of unsigned distance from points sampled on bounding mesh surfaces. We believe that these metrics are stronger than the chamfer distance, which only takes the mean for closest pairs of points.
>
> [1] Sharp, Nicholas, and Alec Jacobson. "Spelunking the deep: Guaranteed queries on general neural implicit surfaces via range analysis." ACM Transactions on Graphics (TOG) 41.4 (2022): 1-16.
>
> [2] Sitzmann, Vincent, et al. "Implicit neural representations with periodic activation functions." Advances in neural information processing systems 33 (2020): 7462-7473.
>
> [3] Müller, Thomas, et al. "Instant neural graphics primitives with a multiresolution hash encoding." ACM transactions on graphics (TOG) 41.4 (2022): 1-15.
>
> [4] Takikawa, Towaki, et al. "Neural geometric level of detail: Real-time rendering with implicit 3d shapes." Proceedings of the IEEE/CVF conference on computer vision and pattern recognition. 2021.
>
> [5] Lindell, David B., et al. "Bacon: Band-limited coordinate networks for multiscale scene representation." Proceedings of the IEEE/CVF conference on computer vision and pattern recognition. 2022.

---

### Official Review · Reviewer_UH7b · 2025-10-31

**Soundness:** 2
**Presentation:** 2
**Contribution:** 2
**Rating:** 2
**Confidence:** 4

**Summary:**

The manuscript follows AdaptiveShell that constructs inner and outer shells which envelope an implicit surface. These shells are rasterized to identify intervals intersecting the surface, and samples are drawn within these intervals for volumetric rendering. Claimed contributions include tightening the inner/outer shells, guaranteeing the enveloping relation, introducing an adaptive split grid for mesh extraction, and demonstrating applications.

**Strengths:**

The high-level idea is reasonable: tighter bounds can reduce wasted samples.
Figures are clear and visually compelling.
The motivation to tighten the bounding shells is sound.

**Weaknesses:**

Writing:
1. There are many equations used to explain the idea, but some appear to be incorrect and could severely impact understanding. For example, in Equation (5) the ≤ symbol likely should be ≥; on line 262 it should be 0, not o; and on line 277 r should be a vector, not a scalar.

Contribution compared with AdaptiveShell:
The method closely follows AdaptiveShell, and the core contributions relative to AdaptiveShell are unclear.
1. If the contribution is ensuring the enveloping relationship between the shells and the implicit surface, then one could refine AdaptiveShell’s shells simply by moving shell grid points along SDF directions. Moreover, the error in Figure 1(b) seems caused by low mesh resolution rather than by SDF erosion/dilation itself.
2. If the contribution is the adaptive split grid, prior work has explored similar ideas, e.g., ACORN [1], so this does not seem to be a core novelty.
3. If the contribution is a tighter bound, Figure 1 shows regions where AdaptiveShell yields a smaller (tighter) bound, such as around the tail.

Experiments:
1. The only main difference from AdaptiveShell is how the inner/outer shells are sampled. This alone does not explain the lower PSNR in Table 2. As an extreme case, reducing the sample density within the interval would lower rendering accuracy (thus PSNR) but also reduce the number of samples, leading to faster rendering.
2. In Figure 4, the number of sample points appears to differ greatly—possibly by more than 10×—which does not align with the reported 3× faster rendering speed.
3. There is no ablation, such as the meshing resolution.

Missing Compared Methods:
Other methods [2-4] that accelerate rendering for implicit representations exist but are not discussed.

[1] ACORN: Adaptive Coordinate Networks for Neural Scene Representation
[2] Gaussian Frosting: Editable Complex Radiance Fields with Real-Time Rendering
[3] Volumetric Rendering with Baked Quadrature Fields
[4] Volumetric Surfaces: Representing Fuzzy Geometries with Layered Meshes

**Questions:**

I could be convinced if the authors address the following:

What are the exact differences from AdaptiveShell, and why are they important? (Ablation required)
How does the method compare with the aforementioned approaches?

---

> ### Author Response · Authors · 2025-11-29
> **Response (1/4)**
>
> Dear Reviewer UH7b,
>
> We appreciate your constructive feedback. The fundamental difference between our method and Adaptive Shells is that our method always outputs **theoretically guaranteed sound bounding meshes regardless of resolution**, while Adaptive Shells provides no such guarantees. The **theoretical soundness** of a pair bounding meshes means that the entire outer mesh must lie strictly outside the implicit surface, and the entire inner mesh lies strictly inside. The soundness guarantee makes sure that there are no missed volumes during rendering or tunneling violations during physics simulation. We address your specific concerns and questions below.
>
> > **Writing:** There are many equations used to explain the idea, but some appear to be incorrect and could severely impact understanding. For example, in Equation (5) the ≤ symbol likely should be ≥; on line 262 it should be 0, not o; and on line 277 r should be a vector, not a scalar.
>
> **Re.:** Thank you for pointing out potential mistakes that may cause confusion. We have corrected the notations in line 262 and 277 in our latest revision.In Equation (5), both $V_+^C$ and $V_-^C$ are supposed to lie on the negative sides of the bounding planes, so the ≤ is not a mistake.

---

> > ### Author Response · Authors · 2025-11-29
> > **Response (4/4)**
> >
> > > **Missing compared methods:** Other methods [2-4] that accelerate rendering for implicit representations exist but are not discussed.
> >
> > **Re.:** We appreciate the reviewer’s observation regarding additional acceleration methods for neural implicit representations. While methods [1- 4] indeed contribute valuable strategies for speeding up rendering, they are largely **complementary rather than directly comparable** to ours. Unlike [1] and [2], GIOM does not modify or intervene in the training process. Method [3] is fundamentally a NeRF-based representation augmented with mesh components, and [4] operates purely on explicit geometry through layered mesh structures. In contrast, GIOM constructs **explicit bounding volumes** and integrates them with **SDFs only at inference time** to enable robust acceleration.
> >
> > [1] ACORN: Adaptive Coordinate Networks for Neural Scene Representation
> >
> > [2] Gaussian Frosting: Editable Complex Radiance Fields with Real-Time Rendering
> >
> > [3] Volumetric Rendering with Baked Quadrature Fields
> >
> > [4] Volumetric Surfaces: Representing Fuzzy Geometries with Layered Meshes

---

> ### Author Response · Authors · 2025-11-29
> **Response (2/4)**
>
> > **Contribution 1**: If the contribution is ensuring the enveloping relationship between the shells and the implicit surface, then one could refine Adaptive Shells simply by moving shell grid points along SDF directions. Moreover, the error in Figure 1(b) seems caused by low mesh resolution rather than by SDF erosion/dilation itself.
>
> **Re.:** We apologize if the presentation of the paper leads to ambiguity or confusion, and would like to restate the main contributions and key novelty here: our approach yields **inherently sound** bounding meshes, and we are able to refine its tightness through adaptive splitting.
>
> Our method is therefore fundamentally different from Adaptive Shells, which cannot guarantee that the bounding meshes do not cross the bounded surface. While one can certainly move vertices along the normal direction to refine Adaptive Shells, tightness would be sacrificed in the process, and there can never be formal guarantee on the conservativeness of the modified bounding meshes: the triangle faces can still intersect with the implicit surface even if all vertices have positive or negative signed distances, not to mention the risk of self-intersection near thin volumes and complex surfaces after mesh expansion or contraction. Increasing the resolution of marching cubes can, of course, reduce approximation errors, but it can still fail on more complex surfaces or when tighter bounds are desired. Unless with infinite memory, one cannot afford converging to a conservative bounding mesh solely by increasing resolution.
>
> Based on your insightful suggestions, we have updated Section 2 (“Certified Bound Extraction and Mesh Enclosures”) and Section 3.2 (“Motivation”) in our latest revision to more clearly distinguish our method from adaptive shell–based approaches and address the concerns raised by the reviewer.
>
> > **Contribution 2:** If the contribution is the adaptive split grid, prior work has explored similar ideas, e.g., ACORN [1], so this does not seem to be a core novelty.
>
> **Re.:** We agree that adaptive spatial partitioning is a broadly explored idea. However, our contribution is not merely splitting adaptively.
>
> (1) **Verification-Driven Splitting with Soundness Guarantee.**
> GIOM splits cells based strictly on the **CROWN-derived SDF bounds**: a voxel is refined **if and only if** its bounds indicate that it may contain the implicit surface. This ensures that every subdivision step is explicitly **driven by soundness**, and the inner/outer envelopes remain a **provably sound enclosure** of the true implicit surface throughout refinement. In contrast, existing adaptive partitioning methods refine based on empirical error or geometric heuristics, and thus do not maintain formal guarantees of soundness.
>
> (2) **Distance-Guided Splitting for Scalable Guarantees.** In addition, we introduce **distance-guided splitting** based on the distance between the inner and outer bounding planes rather than the size of each voxel. This heuristic prioritizes subdivision only where needed for geometric tightness, drastically reducing the number of KD-tree nodes (i.e., verification subproblems).
>
> As shown in the table below, a naïve adaptive split (w/o guidance) leads to near-exponential growth in the number of nodes when the depth increases. In contrast, distance-guided splitting **scales almost linearly**, reducing node count by up to **4x** at deeper levels. This efficiency is crucial for enabling **high-resolution GIOM** without exponential time and memory cost.
> | Tree Depth | 1–17 | 18 | 19 | 20 | 21 | 22 | 23 | 24 | 25 | 26 | 27 |
> |-----------|:----:|---:|----:|-----:|------:|-------:|-------:|--------:|--------:|--------:|--------:|
> | **w/ Distance Guidance**  | same | 7132 | 9560 | 14750 | 21484 | 26130 | 34932 | 40420 | 40446 | 44790 | 32716 |
> | **w/o Distance Guidance** | same | 7132 | 9566 | 14828 | 22356 | 31498 | 49998 | 77284 | 114490 | 185984 | 146808 |
>
>
> > **Contribution 3:** If the contribution is a tighter bound, Figure 1 shows regions where Adaptive Shells yields a smaller (tighter) bound, such as around the tail.
>
> **Re.:** While Adaptive Shells produces tighter bounds at certain high curvature locations, it risks losing soundness, which is always guaranteed by GIOM. Moreover, the same resolution, GIOM often outputs bounds with a **smaller mean unsigned distance**, a metric indicating better overall tightness, as shown in Appendix C (Tables 5–18). We also showcase the overall performance comparison in the response to **Experiments 3.**

---

> ### Author Response · Authors · 2025-11-29
> **Response (3/4)**
>
> > **Experiments 1:** The only main difference from Adaptive Shells is how the inner/outer shells are sampled. This alone does not explain the lower PSNR in Table 2. As an extreme case, reducing the sample density within the interval would lower rendering accuracy (thus PSNR) but also reduce the number of samples, leading to faster rendering.
>
> **Re.:** We would first like to clarify a potential misconception: the inner/outer shells (bounding meshes) are **not sampled** but constructed through a **deterministic process** in both GIOM and Adaptive Shells. We acknowledge the reviewer’s observation on Table 2. The PSNR difference between Adaptive Shells and GIOM is **within 1 dB**, which is not visually perceptible and confirms that GIOM preserves rendering quality. Importantly, both methods use **identical sampling density** along each ray (0.001 units). The small PSNR variance is caused only by minor differences in the ray–surface intersection estimates due to different outer bounds.
>
> Table 2 and Fig. 4(b) already provide a fair comparison at equal per-ray density. However, Adaptive Shells requires more total SDF queries because its looser bounds extend the sampling interval along each ray. To isolate the effect of sampling efficiency, we additionally reduce AdaptiveShell’s sampling density so that both methods consume a **similar total number of samples**. The comparable FPS values reflect this alignment in computational cost.
> Despite this equal-cost setting, AdaptiveShell’s PSNR drops significantly, while GIOM consistently maintains high fidelity, demonstrating that GIOM’s tight bounds lead to **better sample efficiency**:
> |Metric       |  Adaptive Shells ($\delta$=0.01)   |     GIOM  ($\delta$=0.001)
> | ------------| -------:| -----:|
> |PSNR         | 36.46 ± 2.33 | 45.87 ± 5.15 |
> |FPS          | 33.68 ± 10.28 | 30.22 ± 8.81 |
>
> The FPS of GIOM is higher than in the paper because the experiment is run on a new device, but the PSNR is not affected.
>
> Overall, these results confirm that:
> GIOM does not rely on extra samples for high quality
> Adaptive Shells is inherently less sample-efficient due to loose bounds
> > **Experiment 2:** In Figure 4, the number of sample points appears to differ greatly—possibly by more than 10×—which does not align with the reported 3× faster rendering speed.
>
> **Re.:** Number of samples vs. speed-up mismatch: Since our method does not speed up the surface normal calculation and shader application, the final speed up for the entire rendering process is not proportional to the reduction in the total number of samples.
>
> > **Experiments 3:** There is no ablation, such as the meshing resolution.
>
> **Re.:** We have included an ablation study of GIOM on mesh resolution and comparison with Adaptive Shells (dilation/erosion + MC) on seven implicit surfaces in Appendix C (Tables 5–18) in our latest revision. Below we report the average performance across seven objects and compare with Adaptive Shells:
>
> **GIOM (Ours)**
>
> |   Resolution | Time          | Memory      | Mean Distance       |
> |--------------|---------------:|-------------:|---------------------:|
> |      $128^3$ | 53.0 ± 20.5   | 14.8 ± 7.7  | 5.95E-03 ± 3.01E-03 |
> |      $256^3$ | 131.3 ± 73.3  | 46.0 ± 28.0 | 1.60E-03 ± 4.59E-04 |
> |      $512^3$ | 251.6 ± 147.2 | 85.6 ± 51.2 | 9.97E-04 ± 8.30E-05 |
>
> **Adaptive Shells**
>
> |   Resolution | Time         | Memory       | Mean Distance       |
> |--------------|--------------:|--------------:|---------------------:|
> |      $128^3$ | 0.4 ± 0.1    | 1.9 ± 1.0    | 8.25E-03 ± 1.05E-03 |
> |      $256^3$ | 2.5 ± 0.5    | 7.9 ± 4.1    | 4.37E-03 ± 1.41E-03 |
> |      $512^3$ | 18.3 ± 3.6   | 31.6 ± 16.3  | 2.50E-03 ± 1.07E-03 |
> |     $1024^3$ | 166.6 ± 27.3 | 126.9 ± 65.3 | 1.38E-03 ± 7.45E-04 |
>
> To ensure **empirical soundness** for Adaptive Shells, we incrementally adjust the dilation/erosion thicknesses until:
> randomly-sampled points around the outer mesh have strictly positive signed distances, and
>
>
> those around the inner mesh have strictly negative signed distances.
> This ensures no observed violations but provides **no formal guarantees**.
> GIOM, in contrast, enforces **theoretical soundness** for the entire mesh surface. Under the same resolution (e.g., $512^3$), GIOM **improves tightness**, measured by the mean distance from random samples on the bounding meshes to the implicit surface. For similar tightness (e.g., GIOM $512^3$ vs. Adaptive Shells $1024^3$), GIOM **reduces memory cost** despite higher preprocessing time.

---

### Official Review · Reviewer_6vyi · 2025-10-31

**Soundness:** 2
**Presentation:** 3
**Contribution:** 2
**Rating:** 2
**Confidence:** 4

**Summary:**

This paper addresses the problem of extracting meshes from neural implicit surfaces by exploring neural network verification.
The key insight is that deriving formal guarantees for neural signed distance functions (SDFs) can be framed as a verification problem for neural networks [L48].

The proposed method, GIOM, accelerates SDF inference by partitioning the domain into regions using an octree-based bounding scheme. Specifically, GIOM constructs envelopes of the zero-level set through a combination of the CROWN bounding propagation method and voxelized spatial representations.

The method is evaluated on real-time rendering, collision detection, and constructive solid geometry (CSG) operations. However, the motivation for the real-time rendering task could be made clearer.

**Strengths:**

The paper is clearly written and well structured.

Figure 7 shows that GIOM significantly outperforms the CSG-nSDF baseline in terms of reconstruction speed and error, while maintaining comparable query times.

**Weaknesses:**

[L88] Please clarify what is meant by the “specific sound connection” between NN verification and neural implicits.


[L107] Note that 3D Gaussian Splatting (3DGS) does not rely on neural SDFs. Instead, related works such as SuGAR or Gaussian Pull might be more appropriate references.


[L183] It seems GIOM learns two planes per voxel, with the zero-level set lying between them. How large does this octree representation become in practice? It seems potentially expensive, please include memory and speed comparisons.


[L196] Clarify how  M_{-} and M_{+}​ are computed.


[L483] ReLU-based networks may be a limiting factor. That is why all the reconstructions are low resolution piece-wise linear surfaces. This may be a major limitation.

[L767] Additionally, if the proposed verification method is restricted to ReLU networks, it may limit the application of standard INR regularization terms.. For example, the derivative of a ReLU MLP is piecewise constant, preventing the use of the Eikonal constraint, which is commonly used to fit NNs as SDFs (see relevant INR works).

[L108] In the “3D Modeling” section, consider citing dynamic SDF works, e.g.:
Novello et al. “Neural Implicit Surface Evolution.” ICCV 2023.
Yang et al. “Geometry Processing with Neural Fields.” NeurIPS 2021.
Additionally,
Silva et al. “Neural Implicit Mapping via Nested Neighborhoods.” arXiv 2022,
 Is a relevant preprint related to this notion of inner and outer bounds for neural isosurfaces.

The paper “Efficient Neural Network Robustness Certification with General Activation Functions” is listed twice.

Is Figure 2 referenced in the text.

**Questions:**

Why is there no comparison against Spelunking?

How are the neural SDF and its zero-level set shells trained?
Are they optimized jointly, or is the shell construction a post-training step? If separate, how long does the shell computation take?


Why not simply extract the mesh using marching cubes and render it using standard real-time mesh-based pipelines?


In Eq. (3), could you evaluate the neural SDF at the octree vertices and then apply marching cubes for comparison?


The geometric examples are relatively simple. Consider including more complex models (e.g., Lucy, Asian Dragon, Thai Statue from the Stanford 3D Scanning Repository) to better demonstrate scalability and visual quality.

---

> ### Author Response · Authors · 2025-11-29
> **Response (1/3)**
>
> Dear Reviewer 6vyi,
>
> We appreciate your detailed and constructive feedback. The main focus of our method is that our method always outputs **theoretically guaranteed sound bounding meshes regardless of resolutions**. The soundness guarantee makes sure that there are no missed volumes during rendering or tunneling violations during physics simulation. Figure 5 in our latest revision demonstrates a real-world case where the lack of such guarantees causes thin features to disappear in rendering. By increasing branch-and-bound depth, we can also adaptively enhance the tightness of the bounding meshes, which allows for real-time performance at inference time. We address your specific concerns and questions below.
>
> > **W1** [L88] Please clarify what is meant by the “specific sound connection” between NN verification and neural implicits.
>
> **Re.:** We appreciate this comment and have made revisions to link it to the relevant section (3.1 Geometric Queries as NN Verification Problems).
>
> > **W2** [L107], **W7** [L 108], **W8**, **W9** Citations and figure reference issues.
>
> **Re.:** We have updated the related work section to include relevant literature as suggested, removed duplicate papers in the bibliography, and added a reference to Figure 2 in [L249].
>
> > **W3** [L183] Bounding mesh computational cost.
>
> **Re.:** We would first like to clarify a misunderstanding: our method does not “learn” the bounding meshes or the zero level set approximation. They are computed from pre-trained neural SDFs via a deterministic bounding algorithm (CROWN). We have included the time and storage costs of GIOM computation in Appendix C (Tables 5-18) of our latest revision.
>
> > **W4** [L196] Clarify how M_{-} and M_{+} are computed.
>
> **Re.:** The clarification can be found in paragraph **Branch-and-Bound for Voxel Verification** of Section 3.2 Shell Extraction via Guaranteed Bounding Meshes: "In practice, this corresponds to merging all polyhedral meshes derived from voxel
> trimming, yielding the bounding surface meshes $M_-:= ∂V_-$ and $M_+ := ∂V_+$. This final union
> can be performed efficiently with existing mesh processing libraries such as Trimesh [1]. "
>
> > **W5** [L483] and **W6** [L767]: Limited to ReLU-based networks and therefore no Eikonal Regularization.
>
> **Re.:** GIOM is not tied to ReLU networks. We currently use ReLUs because they are well supported by CROWN, but CROWN also supports a wide range of non-linear activations, such as trigonometric, sigmoid, GeLU activations, and etc. Therefore, GIOM does not impose architectural or regularization restrictions on the SDF and remains compatible with a wide range of neural network architectures and computation graphs. In our latest revision, we updated Appendix C with tabular metadata (Tables 5—18) and visualization of bounding meshes on MLPs that included position encoding (trigonometric activation) (Figure 11). The skull and lion statue are encoded with such MLPs, and GIOM has been shown to be able to bound them tightly.

---

> ### Author Response · Authors · 2025-11-29
> **Response (2/3)**
>
> > **Q1** Why no comparison against Spelunking the Deep?
>
> **Re.:** We have already compared with the interval tracing method proposed by Spelunking the Deep in the original submission (Section 4 **Real-Time Rendering** paragraph and Table 2 second row). By using pre-computed bound meshes, we avoid interval computation on the fly to achieve better efficiency in rendering. We have clarified this point in our latest revision to avoid confusion.
>
> > **Q2** Is GIOM(-Z) post-training and how long does its construction take?
>
> **Re.:** Our bounding meshes and zero-level set construction are strictly **post-hoc** and **training-free**. We added detailed per-object construction statistics in Appendix C (Tables 5–18). Below we report the average performance across seven objects and compare with Adaptive Shells (dilation/erosion + MC):
>
> **GIOM (Ours)**
>
> |   Resolution | Time (s)         | Memory (MB)      | Mean Distance       |
> |--------------|---------------:|-------------:|---------------------:|
> |      $128^3$ | 53.0 ± 20.5   | 14.8 ± 7.7  | 5.95E-03 ± 3.01E-03 |
> |      $256^3$ | 131.3 ± 73.3  | 46.0 ± 28.0 | 1.60E-03 ± 4.59E-04 |
> |      $512^3$ | 251.6 ± 147.2 | 85.6 ± 51.2 | 9.97E-04 ± 8.30E-05 |
>
> **Adaptive Shells**
>
> |   Resolution | Time (s)        | Memory (MB)      | Mean Distance       |
> |--------------|--------------:|--------------:|---------------------:|
> |      $128^3$ | 0.4 ± 0.1    | 1.9 ± 1.0    | 8.25E-03 ± 1.05E-03 |
> |      $256^3$ | 2.5 ± 0.5    | 7.9 ± 4.1    | 4.37E-03 ± 1.41E-03 |
> |      $512^3$ | 18.3 ± 3.6   | 31.6 ± 16.3  | 2.50E-03 ± 1.07E-03 |
> |     $1024^3$ | 166.6 ± 27.3 | 126.9 ± 65.3 | 1.38E-03 ± 7.45E-04 |
>
> To ensure **empirical soundness** for Adaptive Shells, we incrementally adjust the dilation/erosion thicknesses until:
> 1. randomly-sampled points around the outer mesh have strictly positive signed distances, and
> 2. those around the inner mesh have strictly negative signed distances.
>
> This ensures no observed violations but provides **no formal guarantees**.
> GIOM, in contrast, enforces **theoretical soundness** for the entire mesh surface. Under the same resolution (e.g., $512^3$), GIOM **improves tightness**, measured by the mean distance from random samples on the bounding meshes to the implicit surface. For similar tightness (e.g., GIOM $512^3$ vs. Adaptive Shells $1024^3$), GIOM **reduces memory cost** despite higher preprocessing time. GIOM-Z takes approximately half of the time as GIOM as it requires building only one layer of mesh instead of two.

---

> ### Author Response · Authors · 2025-11-29
> **Response (3/3)**
>
> > **Q3** Why not extract surface mesh via marching cubes and render it directly?
>
> **Re.:** We have shown that directly extracting the surface mesh with marching cubes can lead to artifacts and missing volumes for thin objects in Fig. 5 of our latest revision. We demonstrate how our method yields robust results compared to marching cubes in the same figure. Additionally, we show that simply rendering the surface mesh extracted via marching cubes leads to low PSNR compared to other methods in Table 2. While one can further argue that increasing the resolution of marching cubes can reduce artifacts or missing volumes, it would require human supervision and probably a higher memory budget in practice, which is less preferred than a method with automatic precision guarantees like GIOM.
>
> > **Q4** Could you evaluate the neural SDF at the octree vertices and then apply marching cubes for comparison?
>
> **Re.:** Evaluating the SDF at cube vertices and then running marching cubes yields the same mesh as directly running marching cubes. The only difference is that at high resolution, the former reduces mesh extraction time.
>
> To contextualize this, we provide below a comparison between our method (GIOM), standard CPU marching cubes, and Spelunking the Deep’s GPU+Octree implementation:
> | Resolution | GIOM (CPU mesh) | CPU MC | GPU+Octree MC |
> | ----------| ---------------:| ------:| ---------:|
> | $128^3$      | 43 s            | 0.3 s  | 0.2 s     |
> | $256^3$      | 94 s            | 2 s    | 1 s       |
> | $512^3$      | 170 s           | 15 s   | 4 s       |
> | $1024^3$    | 264 s           | 157 s  | 12 s      |
>
> In Appendix C Tables 5—18 of our latest revision, we compare GIOM with CPU-based marching cubes because our mesh-construction stage is also CPU-based. While our voxel verification and subdivision run on the GPU, the final mesh extraction currently relies on a CPU pipeline. Comparing against GPU-accelerated MC methods would therefore mix algorithmic differences with hardware differences and obscure the fairness of the comparison.
>
> Our focus is on guaranteed bounding mesh construction, rather than optimizing low-level geometry extraction kernels. Importantly, mesh extraction is a **one-time preprocessing cost**: once the bounding mesh is constructed for a network, it can be reused for all subsequent real-time rendering or simulation queries without re-computation. Therefore, our method remains practical even without GPU-optimized extraction. That said, integrating GPU-compatible mesh construction is an exciting future engineering direction that can further reduce this one-time cost, and we plan to explore such extensions in future work.
>
> > **Q5** The geometric examples are relatively simple. Consider including more complex models to better demonstrate scalability and visual quality.
>
> **Re.:** We have updated Appendix C (Tables 13–18 and Figure 11) in our latest revision to include more complex shapes, such as the skull (with holes), lion statue (uneven surface features), and scorpion (thin, long structures). We fitted the new shapes with MLPs with 8 layers and 64 neurons in each hidden layer. For the skull and lion statue, we added positional encoding with 10 frequencies (from $2^-3$ to $2^6$) to enhance neural SDF accuracy. We list GIOM performance at resolution $512^3$ for all implicit surfaces included in the paper in the ascending order of visual complexity to demonstrate scalability.
>
> | Object      |   Time (s) |   Memory (MB) |   Mean Distance |
> |-------------|--------:|----------:|-----------------:|
> | Koala       |  124.4 |     37.8 |       1.15E-03  |
> | Cat         |  100.1 |     29.9 |      1.01E-03 |
> | Fox         |  159.5 |     46.5 |       8.91E-04 |
> | Tree        |  299.1 |     90.2 |       1.01E-03  |
> | **Scorpion**    |  320.6 |    112.0   |       9.25E-04 |
> | **Lion Statue** |  232.2 |    110.3 |       9.75E-04 |
> | **Skull**       |  525.3 |    172.2 |       1.00E-03 |
>
> For complex objects, GIOM is still tight (with low mean distance) under low memory budgets compared to Adaptive Shells despite being more time consuming. We show the performance of Adaptive Shells at resolution $1024^3$ with comparable tightness below for reference.
>
> | Object      |   Time (s) |   Memory (MB) |   Mean Distance |
> |-------------|--------:|----------:|-----------------:|
> | Koala       |  175.8 |    125.0   |      9.99E-04  |
> | Cat         |  126.9 |     67.7 |       9.98E-04  |
> | Fox         |  131.2 |     63.9 |       1.00E-03  |
> | Tree        |  198.7 |     89.4 |       3.00E-03     |
> | **Scorpion**    |  186.2 |    140.7 |       1.00E-03  |
> | **Lion Statue** |  191.7 |    115.9 |       9.95E-04 |
> | **Skull**       |  151.5 |    269.8 |       9.99E-04 |
>
>
> [1] Dawson-Haggerty et al. 2019. trimesh. https://trimesh.org/. Version 3.2.0.
>
> [2] Efficient Neural Network Robustness Certification with General Activation Functions
>
> [3] Automatic Perturbation Analysis for Scalable Certified Robustness and Beyond

---

### Note · Authors · 2026-01-11

**Comment:**

The authors have decided to withdraw this submission.

**Withdrawal Confirmation:**

I have read and agree with the venue's withdrawal policy on behalf of myself and my co-authors.